# Physiological and genetic characterization of heat stress effects in a common bean RIL population

Yulieth Vargas, Victor Manuel Mayor-Duran[¤a], Hector Fabio Buendia, Henry Ruiz-Guzman[¤b], Bodo Raatz [ID] *

International Center for Tropical Agriculture (CIAT), Cali, Colombia

¤a Current address: Semillas Valle, Yumbo, Colombia
¤b Current address: Texas A&M University, TX, United States of America
* bodo81@hotmail.com

## Abstract

Heat stress is a major abiotic stress factor reducing crop productivity and climate change models predict increasing temperatures in many production regions. Common bean (*Phaseolus vulgaris* L.) is an important crop for food security in the tropics and heat stress is expected to cause increasing yield losses. To study physiological responses and to characterize the genetics of heat stress tolerance, we evaluated the recombinant inbred line (RIL) population IJR (Indeterminate Jamaica Red) x AFR298 of the Andean gene pool. Heat stress (HS) conditions in the field affected many traits across the reproductive phase. High nighttime temperatures appeared to have larger effects than maximum daytime temperatures. Yield was reduced compared to non-stress conditions by 37% and 26% in 2016 and 2017 seasons, respectively. The image analysis tool HYRBEAN was developed to evaluate pollen viability (PolVia). A significant reduction of PolVia was observed in HS and higher viability was correlated with yield only under stress conditions. In susceptible lines the reproductive phase was extended and defects in the initiation of seed, seed fill and seed formation were identified reducing grain quality. Higher yields under HS were correlated with early flowering, high pollen viability and effective seed filling. Quantitative trait loci (QTL) analysis revealed a QTL for both pod harvest index and PolVia on chromosome Pv05, for which the more heat tolerant parent IJR contributed the positive allele. Also, on chromosome Pv08 a QTL from IJR improved PolVia and the yield component pods per plant. HS affected several traits during the whole reproductive development, from floral induction to grain quality traits, indicating a general heat perception affecting many reproductive processes. Identification of tolerant germplasm, indicator traits for heat tolerance and molecular tools will help to breed heat tolerant varieties to face future climate change effects.

## Introduction

Common bean (*Phaseolus vulgaris* L.) is the most important grain legume for direct human consumption [1]. Its production covers diverse areas, being cultivated all over the world [2].

**Data Availability Statement:** Additional data is available from Harvard Dataverse: https://doi.org/ 10.7910/DVN/KXIDBW.

**Funding:** BR - The work was funded by Tropical Legumes III - Improving Livelihoods for Smallholder Farmers: Enhanced Grain Legume Productivity and Production in Sub-Saharan Africa and South Asia (OPP1114827). Funding agency: Bill & Melinda Gates Foundation. www. gatesfoundation.org BR - We would like to thank USAID for their contributions through the CGIAR Research Program on Grain Legumes and Dryland Cereals Funding agency: USAID www.usaid.gov BR - We thank USAID for funding in the form of the "Global Hunger and Food Security Research Strategy: Climate Resilience, Nutrition, and Policy – Feed the Future Innovation Lab for Climate Resilience in Beans (CRIB)" Project (#AID--OAA--A--13--OO077). Funding agency: USAID www. usaid.gov The funders had no role in study design, data collection and analysis, decision to publish, or preparation of the manuscript.

**Competing interests:** The authors have declared that no competing interests exist.

World production exceeds 30 million metric tons per year, mainly in Asia, Latin America and Africa [3]. Common bean has a large ecological and geographic range of adaptation, from tropical moist highlands to hot low altitudes and dry conditions [4, 5].

Climate change leads to problems of heat stress and drought, which pose a threat to agricultural production [6–8]. Heat stress is a major constraint for crop production worldwide and it is expected to have a growing impact on food security due to climate change [9]. Current research suggests that the average global temperatures have increased by ~0.8˚C since 1880 [10]. Most of that increase appeared in recent times, with two-thirds of the warming occurring since 1975, at a rate of roughly 0.15–0.20˚C per decade [11]. The elevated temperatures will negatively affect the growth and yield of various crops, particularly in tropical and subtropical regions [12]. Developing countries situated in these regions will suffer even more from these changes, as they lack the resources to react to changing conditions. Large parts of current common bean growing areas in southeastern Africa are predicted to become unsuitable for bean cultivation by the year 2050 [13, 14].

Heat stress is observed when plants are exposed to elevated temperatures for a period long enough to cause negative effects on growth, development and productivity of plants [15]. High nighttime temperatures during the reproductive phase have been reported to cause heat stress in common bean, and to a lesser degree, high daytime temperatures [16]. These higher temperatures lead to the abortion of flowers, buds and pods, reduce pollen viability and anther dehiscence, cause damage to pollen tube formation, and reduce seed filling and seed size, thereby resulting in significant yield reduction [15, 17, 18]. Temperatures higher than 20˚C during the night or higher than 30˚C during the day have been reported to cause seed yield reduction [19]. Common bean genotypes of the Andean gene pool are commonly grown at mid to mid-high altitudes (1400–2800 masl) or in cooler climates, whereas genotypes of the Mesoamerican gene pool adapt to low to mid altitude ranges (400–2000 masl) with higher temperatures. For this reason, Andean beans are expected to be more sensitive to high temperatures [4].

Genetic improvement for heat stress in common beans is carried out through selection of germplasm under stress conditions in the field or in controlled environments. Yield and related traits such as pollen viability are used to select superior genotypes [17, 18, 20, 21]. At the International Center for Tropical Agriculture (CIAT) climbing beans with improved adaptation to heat environments were developed [20].

Genetic mapping of QTLs is a method to identify genetic loci underlying variation in phenotypic traits of interest. QTL for many traits have been mapped in common bean, and linked markers have been utilized for marker-assisted selection [22]. Single nucleotide polymorphism (SNP) have become the preferred marker type, because they are the most abundant type of DNA polymorphism. SNP based markers have been recently used to identify QTLs in bean associated with other types of abiotic stress such as drought [21–24].

This study aimed to investigate the effects of heat stress on the Andean RIL population of IJR x AFR298. Physiological traits (indicators) were evaluated to identify traits that were affected by heat stress and those associated with superior agronomic performance under heat stress. QTL mapping was performed to identify genomic regions associated with heat tolerance traits and to identify molecular markers that may be used in MAS to generate heat tolerant lines.

## Materials and methods

### Plant material

The recombinant inbred lines (RIL) population IJR x AFR298 with a total of 107 lines was developed crossing the two parental lines from the Andean gene pool and advancing F2 lines

by single seed descent up to F6. The genotype IJR (Indeterminate Jamaica Red) is an indeterminate bush type with pink /red mottled seed color. Different studies have reported it as tolerant to heat stress [20, 25]. AFR298 is a CIAT breeding line, released in Colombia under the name ICA Quimbaya. It is a red seeded determinate bush type which shows broad adaptation to many different growth environments.

## Experimental locations

The evaluations of heat stress (HS) and non-stress (NS) trials were conducted in the field at two locations. The first location was under high temperature conditions on the farm La Santica, at Alvarado, Tolima, Colombia (04˚32' 45.52" N, 74˚ 57' 39.56", elevation of 460 masl), where two plantings were conducted in dry seasons between July and October of 2016 and 2017. The second location that was considered a NS treatment was located at the CIAT experimental station at Palmira, Valle del Cauca (03˚30' 20.39" N, 76˚20' 28.13", elevation of 973 masl). The trial at this location was conducted between the months of July and October of 2017. Phenology data was taken from unreplicated seed multiplication plots at the same location of Palmira under NS conditions in 2018. Weather data in Alvarado was logged with a meteorological station HOBO® onset U30 (Onset Computer Corporation, Bourne, MA, USA) located in the experimental field. For the Palmira location the CIAT meteorological station recorded weather data. Temperature profiles during trials are shown in S1 Fig.

## Experimental design

For the evaluation of HS2016 in Alvarado, an 10x12 alpha lattice design was used, which was constituted of 10 blocks of 12 individuals and three repetitions. The experimental unit was one row of 3.72 meters for each repetition, at a distance of 0.60 m between rows, with a sowing density of 7.5 cm between plants. For HS2017, also one row of 3.72 meters per line was sown. For this trial only one replicate was sown. For the evaluations three plants from each experimental plot were used as biological replicates. In this trial no data from the lines RIS 38, RIS 44, RIS 46, RIS 50 and RIS 51 was collected due to poor germination.

For the NS trial (NS2017) in Palmira, an 14x8 alpha lattice design was used, which was constituted of 14 blocks of eight individuals and three repetitions. The experimental unit was one row of 3.72 meters for each line by repetition, at a distance between rows of 0.60 m, with a sowing density of 7.5 cm between plants.

Standard agronomic practices were applied over the plant growing seasons across trials, including irrigation, the application of fungicide, seed treatment and foliar insecticides when necessary.

## Phenotypic evaluations

**Evaluations during flowering period.** For evaluations during flowering period the following traits were recorded:

**Days to flowering (DF):** Number of days after planting until 50% of the plants have at least one open flower.

**Pollen viability (PolVia):** The percentage of pollen viability was estimated following the protocol proposed by Polanía et al [26]. One day before anthesis between six and eight floral buds were collected randomly per line between 06:00 and 08:30 am. They were stored in previously labeled plastic jars, which contained a solution of 3:1 96% alcohol—glacial acetic acid, and preserved at a temperature of 4˚ C until evaluation. With the help of a stereoscope the anthers were removed from the buds and placed on a microscope slide, and pressure was applied to extract the pollen grains. A drop of 1% acetocarmine (Sigma-Aldrich) stain was

added and the coverslip was placed under a microscope. Three floral buds per line were evaluated. The color of pollen grains was evaluated through photographic records obtained with the AxioCam ERc5s (ZEISS) camera attached to the microscope with the 4X objective (S2 Fig). Pollen grains were counted and classified into viable and non-viable by using the software HYRBEAN (described below).

**Indehiscent anthers (IA):** To determine the percentage of floral buds with indehiscent anthers, three buds per line were evaluated. In a stereoscope the anthers of the buds were removed and placed on a slide. Indehiscent anthers were considered those that did not easily release pollen after pressure was applied and the anthers that liberated the pollen without difficulty were considered dehiscent (Fig 1).

**Evaluations at harvest time.** For phenotypic evaluations at harvest, five plants were taken per line and repetition in the HS2016 and NS2017 trials, and three plants per line in the HS2017 trial. The traits evaluated were the following:

**Yield per ha (Yd):** The estimation of the yield extrapolated to kilograms per hectare was made for each of the plots.

**Seed weight per plant (SdWPl):** Weight in grams of seeds harvested from each plant.
**Number of seeds per plant (SdPl):** Total number of seeds harvested per plant.
**Pod weight per plant (PdWPl):** Weight in grams of pods harvested for each plant.
**Number of seeds per pod (SdPd):** Average number of seeds formed per pod.
**Number of pods per plant (PdPl):** Total number of pods harvested per plant.
**100 seed weight (100SdW):** Corresponds to the weight in grams of 100 seeds of each sample.

**Pod harvest index (PHI):** PHI in % is calculated as (seed biomass as dry weight at harvest) / (pod biomass as dry weight at harvest) x 100 [27].

**Classification of pods:** The pods were classified based on the seed characteristics as follows (Fig 2):

1. **Pods with well-formed seeds (PdWF):** Percentage of pods with well-formed seed.

2. **Pods with shriveled seeds (PdShr):** Percentage of pods with partial seed filling. These seeds have a deformed or shriveled appearance and of different sizes.

3. **Pods with partially well-formed seeds (PdPWF):** Percentage of pods with one or more locules with aborted embryos and the remaining locules may have well-formed or shriveled seeds.

4. **Pods with aborted embryos (PdAE):** Percentage of pods with aborted embryos. These pods have a normal size, yet appear empty or show undeveloped seeds.

5. **Pods that are empty (PdE):** Percentage of pods that are empty. These pods are small in size without seed filling (aborted ovules).

## HYRBEAN, a computer vision tool to evaluate pollen viability

After pollen grain staining, viable and non-viable grains are usually counted on a microscope slides (Fig 3A, S2 Fig). To replace this time-consuming, and potentially biased manual process, and for more accurate and cost-effective pollen phenotyping, the novel image-based open-source software HYRBEAN (interface implementing OpenCV for image processing and custom algorithms, available under https://github.com/haruiz/HYRBEAN) was developed and implemented. The tool uses machine learning and image processing techniques to count and classify the pollen grains on the images collected by a 2d camera attached to the microscope.

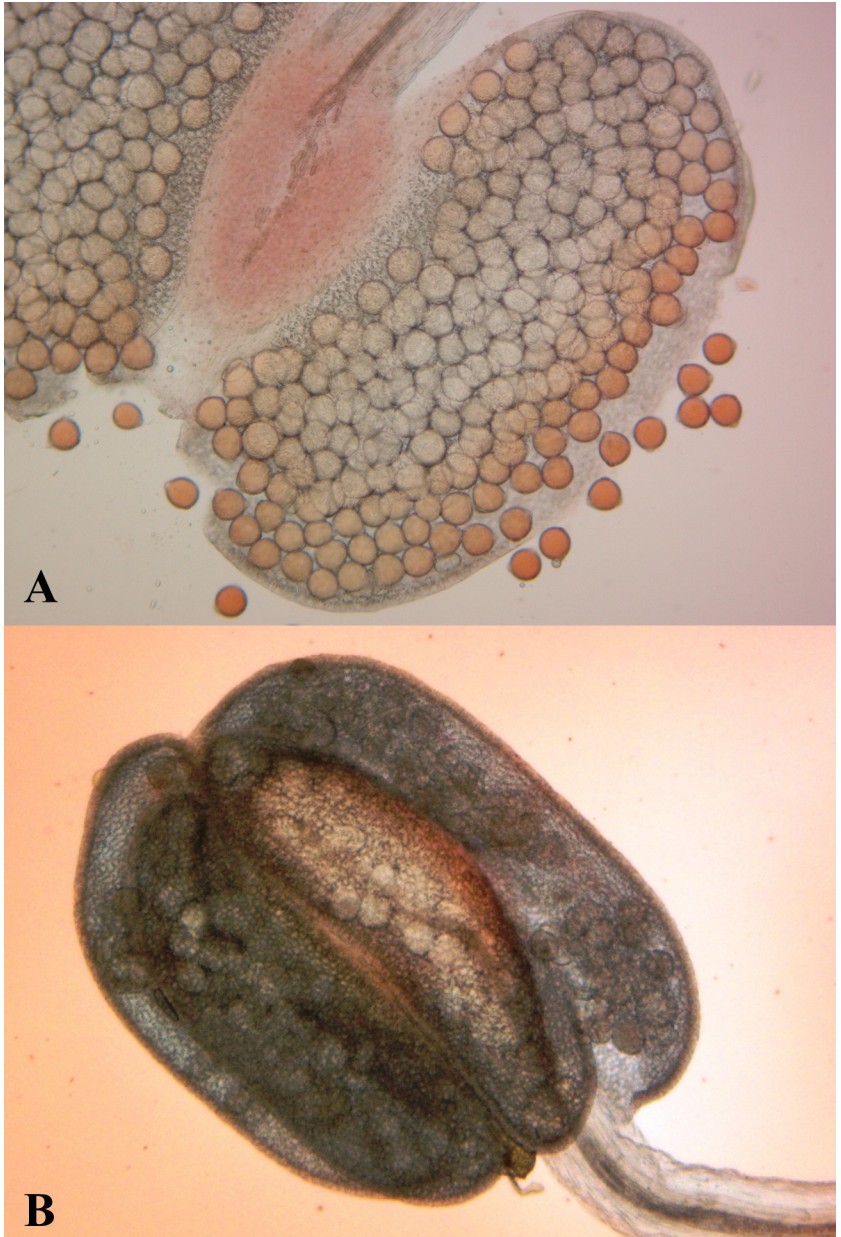

**Fig 1. Evaluation of anther indehiscence in common bean.** A: Dehiscent anthers, B: Indehiscent anthers.

HYRBEAN image analysis workflow works as follows. Image segmentation to identify pollen grains was achieved with the following steps using OpenCV software package [28]. a) each input RGB image is converted to grayscale; b) then, the Sobel filter in x, y directions is employed for highlighting the pollen grains present on the image; c) to detect connected areas, the canny filter followed by a close morphological operation are applied; d) sequentially, the watershed segmentation algorithm is applied to separate the touching regions; e) finally, the contours are extracted utilizing the find contours function with the OpenCV API. Only the contours where the circularity and area have higher significance scores than 0.5 and 100, respectively, are considered. The results of the segmentation can be observed in Fig 3B.

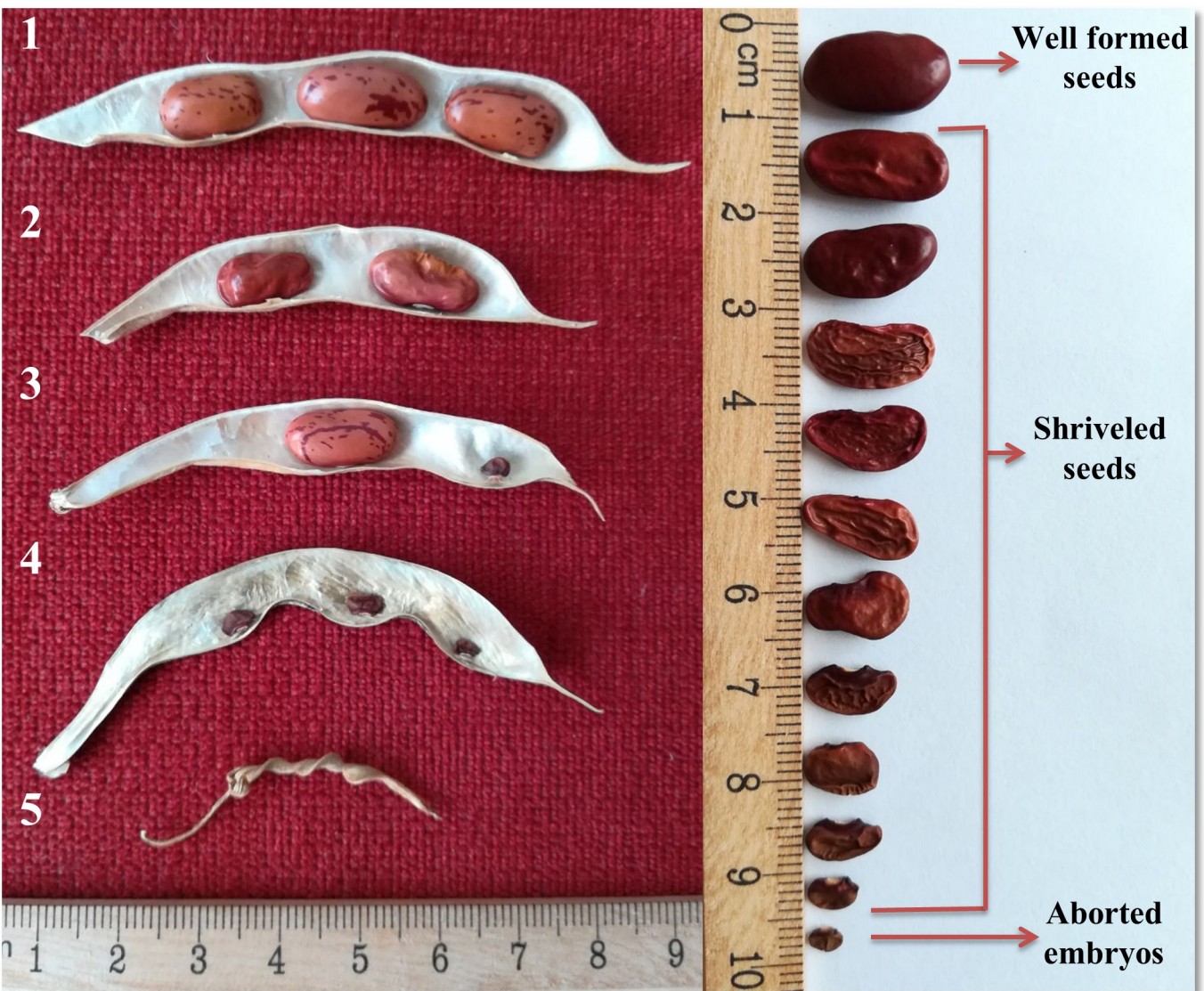

**Fig 2. Classification of pods according to the seed characteristics.** 1: Pods with well-formed seeds (PdWF); 2: Pods with shriveled seeds (PdShr); 3: Pods with partially well-formed seeds (PdPWF); 4: Pods with aborted embryos (PdAE); 5: Pods that are empty (PdE). On the right the range of seed formation defects that was observed in the heat stress trials is shown.

After the pollen segmentation, a dataset with 19 features extracted using image processing is created to train the machine learning classifier. Ten of them correspond to geometric attributes that carry information regarding the dimension and shape. Two are texture related, providing information about how pixel intensities are distributed on the image, and the others are associated with the color content from two different color spaces, HSV and RGB, respectively. Most features were explored in [29]. The complete list is described in the S1 Table.

Differences between the viable and non-viable pollen grains are easily perceived visually (Fig 3A). Thus, in this study, a support vector machine (SVM) model [30] with a linear kernel was trained with around 2000 images by class and 500 iterations to make the classification, using proportions of 70% as training and 30% as validation set. The model was capable of distinguishing between viable pollen and non-viable pollen (Fig 3B), with a Pearson correlation

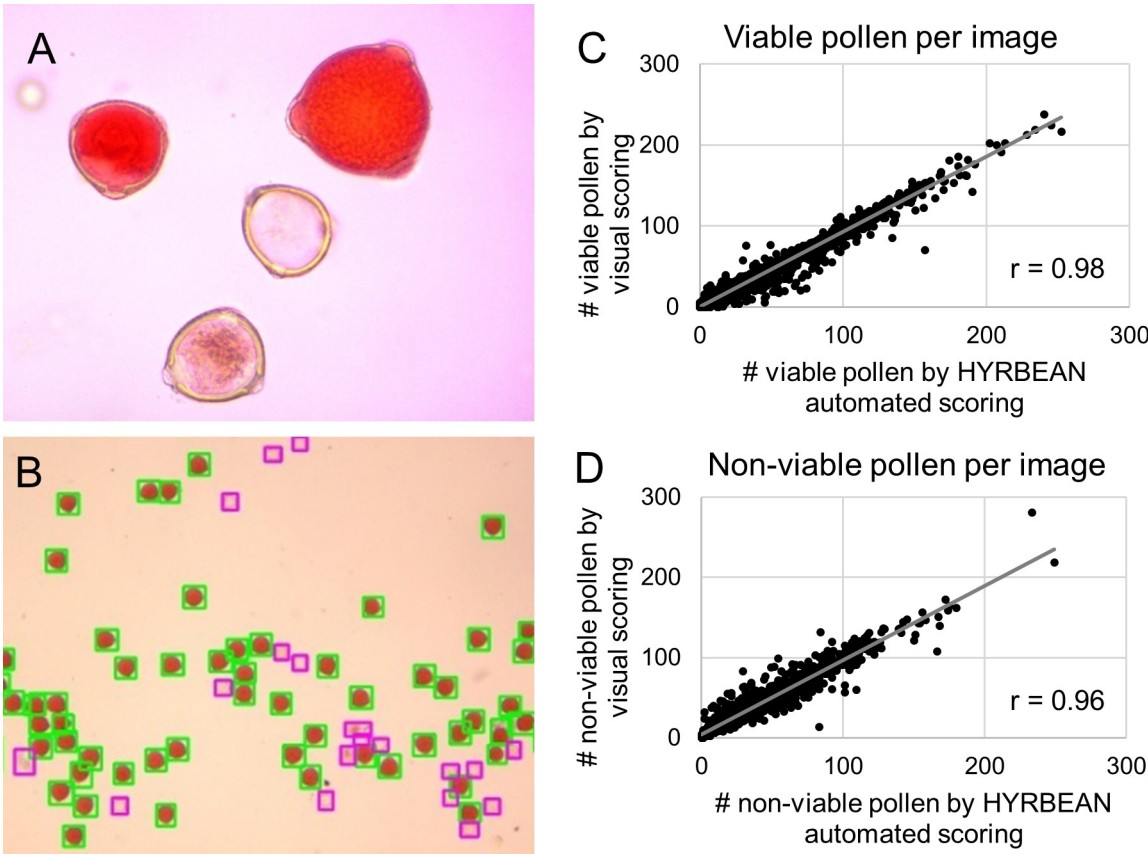

**Fig 3. Detection of viable and non-viable pollen.** (A) Visual differences between viable (stained red) and non-viable (transparent) pollen stained with acetocarmine. (B) Excerpt of a pollen grain image showing segmentation and pollen classification results from HYRBEAN image analysis tool. Viable pollen is boxed in yellow, non-viable pollen in blue. (C, D) Correlations between pollen counts from visual scoring and HYRBEAN automated scoring, for viable and non-viable pollen, based on 2000 images from acetocarmine pollen staining. r represents the Pearson correlation.

of 0.98 and 0.96, respectively, between visual and automated pollen counts of viable and non-viable pollen (Fig 3C and 3D).

## Phenotypic data analysis

The means of the trials HS2016 and NS2017 were obtained through a Best Linear Unbiased Estimation (BLUE) analysis, considering genotypes as fixed effects. Analyses were performed using the programs Plant Breeding Tools V1.4 [31] and META-R V6.01 [32]. For the HS2017 trial the arithmetic mean obtained from the three observations per plot was used. Additionally, trial data under stress was combined and termed heat stress combined (HSC) from years HS2016 and HS2017 generating BLUEs, taking the one repetition of the HS2017 trial as a fourth repetition in addition to the three repetitions from HS2016. Pearson's correlations were calculated between trials and between traits, using the statistical software R [33], and visualized with the package ggplot2 [34]. The Tukey test was also calculated using the same software to compare the means of traits. Yield retention under heat stress (%Yd in HS), was calculated as $Yd_{HS} / Yd_{NS} *100$, and percentage of days to flowering in heat stress (%DF in HS) alike, as $DF_{HS} / DF_{NS} * 100$.

Phenotypic and genotypic data is available on dataverse following the link: https://doi.org/10.7910/DVN/KXIDBW.

## DNA extraction and genotyping

For DNA extraction leaf tissue from young leaves was collected pooling four F7 plants. For each line tissue samples from four individual seedlings were pooled. DNA extraction was carried out according to a urea buffer protocol including a phenol extraction reported by Chen et al [35]. DNA quality was verified on 1% agarose gels. The samples were sent to USDA–ARS Beltsville for genotyping with a set of 5,398SNPs using BARCBean6K_3 SNP BeadChip [36].

## Genetic mapping and identification of QTL

Genotyping generated a total of 5398 markers of which only 400 were polymorphic between parental lines. Markers that were redundant were removed with the BIN function (Binning of redundant markers) of the IciMapping QTL V 4.1 program resulting in 162 markers remaining [37].

The genetic map was constructed using IciMapping QTL, using the Kosambi map function to calculate the genetic distance in centiMorgans (cM) between markers; SARF was used as criteria for ordering and ripple. The line RIS39 was not included in the analysis due to genotyping errors.

The QTL identification was conducted using the composite interval mapping analysis (CIM) of the IcIMapping QTL software. The thresholds for the QTLs were determined by the generation of 1000 permutations. The genomic regions that proved to be significant in the analysis were visualized using MapChart [38].

## Results

### Heat stress affects reproductive development

Heat stress (HS) effects were evaluated in two field trials in Alvarado, Colombia 2016 and 2017, and compared to a non-stressed (NS) trial in Palmira, Colombia 2017. Average minimum and maximum daily temperatures were consistently higher in HS trials, 3–7°C above the NS trial (Fig 4). HS2016 presented higher temperatures throughout the vegetative phase compared to HS2017. During the reproductive phase the lowest minimum daily temperatures (nighttime temperatures) were registered in HS2016, whereas greatest maximum temperatures (daytime) were observed in HS2017.

Evaluating 16 traits related to flower and seed development, we identified 11 traits that showed a significantly different phenotypic response under HS compared to NS conditions (Table 1). HS2016 and NS2017 represent higher quality data sets compared to the HS2017 trial, which was not replicated, hence, in nearly all trials standard deviations are higher compared to HS2016 or NS trials. Onset of flowering was noted about five days later in the HS trials than in NS conditions. HS reduced pollen viability significantly, from 90% in NS to 75% in both HS trials. Indehiscent anthers evaluated in 2017 were observed under HS at a frequency of 6% and barely noted in NS (0.5%).

Next to the flowering phase, significant effects of heat stress were also observed at harvest time. Mean yields in NS were 1.5 t/ha, whereas under HS yields were reduced by 37% and 26% in 2016 and 2017, respectively. Also, other yield components were found reduced in both HS trials compared to NS conditions. Mean values under HS for SdWPl were lowered 32%, SdPl by 15%, PdWPl by 24% and SdPd by 37%. Parental line IJR which was considered more heat

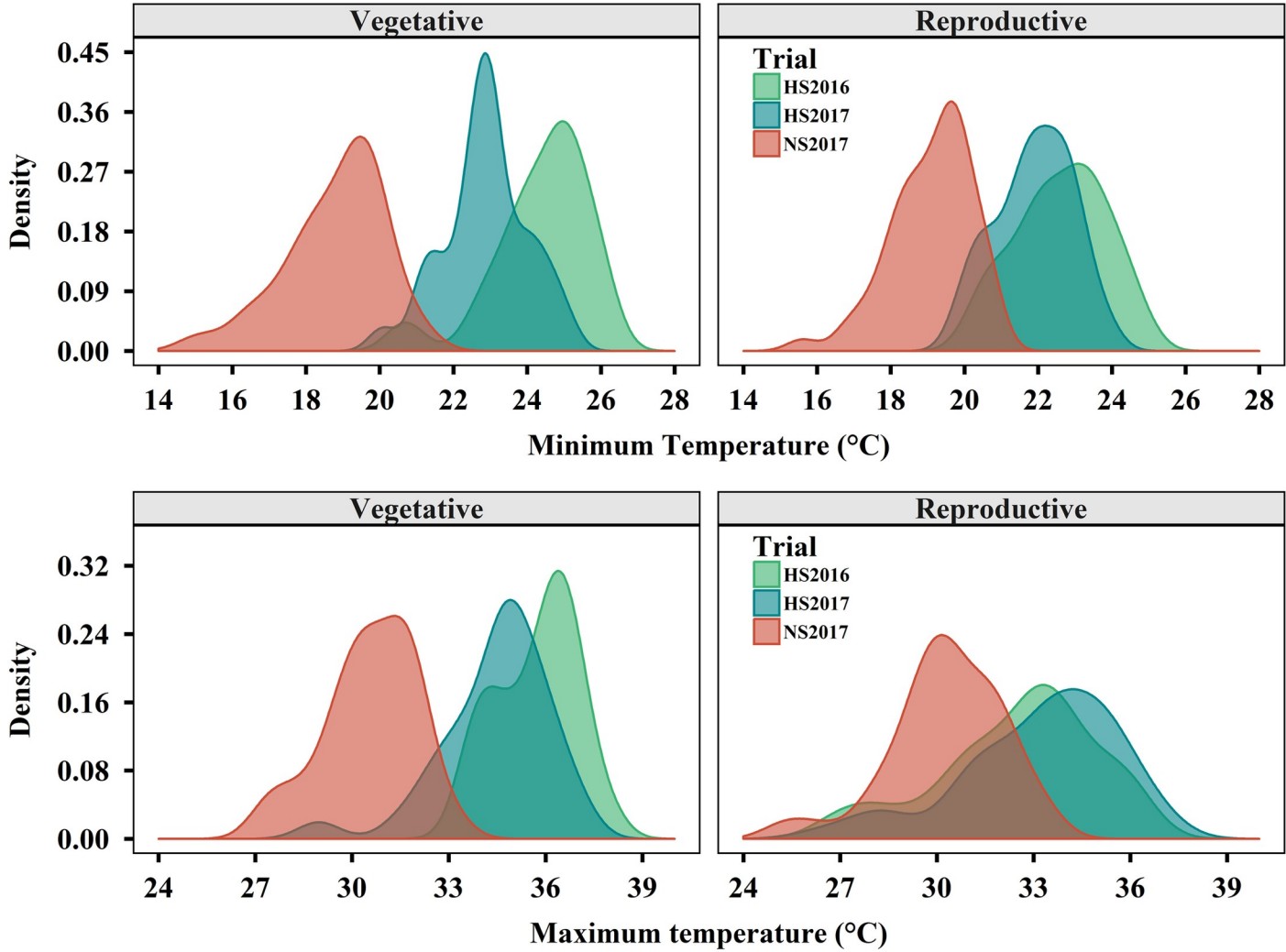

**Fig 4. Distributions of maximum and minimum daily temperatures during vegetative and reproductive phases.** Shown are the heat stress trials (HS) in Alvarado 2016/2017 and non-stress trial (NS) in Palmira 2017.

tolerant showed better performance under HS with 68% more yield than parental line AFR298 (S3 Fig). Overall, weight of harvested pods and grain was higher in NS.

In contrast, PdPl was found to be higher in both HS trials, the increase was significant in HS2016. This points towards defects in reproductive development under HS, where pods are not filled with grain, and consequently more pods continue to be formed. At harvest several further effects on grain quality were observed. 100SdW was significantly reduced by 26% in HS2016, the trial which experienced highest nighttime temperatures. Similarly, PHI was found reduced in stronger HS conditions HS2016 while no significant differences were observed between NS and HS2017.

A pod classification system was introduced to characterize seed formation defects (Fig 2). Under NS highest fraction of pods with well-formed seed without defects (PdWF) was observed. Similarly pods with partial defects (PdPWF) were higher in NS. On the other hand, pods with shriveled seeds were found in much higher proportions in HS trials, increased by factor ten and four in both HS trials, respectively. Likewise, pods with embryos that failed to develop (PdAE) were observed mostly in HS2016, and complete seed development failure

**Table 1. Overview of phenotypic data of the IJR x AFR298 population evaluated in heat stress (HS) and non-stress (NS) trials.**

| Trait | Trial | Mean | Minimum | Maximum |
|---|---|---|---|---|
| **Evaluated at flowering** | | | | |
| Days to flowering (DF) | NS2018 | $33.87 \pm 1.62^b$ | 30.00 | 37.00 |
| | HS2016 | $38.46 \pm 2.42^a$ | 33.02 | 42.98 |
| | HS2017 | $38.07 \pm 4.20^a$ | 33.00 | 50.00 |
| | HSC | $38.37 \pm 2.39^a$ | 34.11 | 44.25 |
| Pollen viability (PolVia) | NS2017 | $90.57 \pm 7.46^a$ | 29.86 | 97.25 |
| | HS2016 | $75.16 \pm 13.81^b$ | 8.77 | 96.72 |
| | HS2017 | $74.95 \pm 12.59^b$ | 34.92 | 94.17 |
| | HSC | $75.24 \pm 11.67^b$ | 25.33 | 95.69 |
| Indehiscent anthers (IA) | NS2017 | $0.51 \pm 4.38^b$ | 0 | 44.44 |
| | HS2017 | $6.08 \pm 13.98^a$ | 0 | 77.78 |
| **Evaluated at harvest** | | | | |
| Yield per ha (Yd) | NS2017 | $1,512.5 \pm 427.1^a$ | 725.71 | 3101.67 |
| | HS2016 | $954.2 \pm 437.9^b$ | 3.85 | 1887,86 |
| | HS2017 | $1,112.4 \pm 665.2^b$ | 32.50 | 3225.00 |
| | HSC | $986.7 \pm 406.0^{\ b}$ | 87.44 | 1862.65 |
| Seed weight per plant (SdWPl) | NS2017 | $9.41 \pm 3.13^a$ | 3.76 | 19.30 |
| | HS2016 | $6.10 \pm 2.58^c$ | 0.84 | 14.02 |
| | HS2017 | $7.42 \pm 4.43^b$ | 0.22 | 21.50 |
| | HSC | $6.41 \pm 2.47^{bc}$ | 1.15 | 13.73 |
| Number of seed per plant (SdPl) | NS2017 | $27.26 \pm 8.55^a$ | 10.33 | 50.17 |
| | HS2016 | $23.76 \pm 9.63^b$ | 3.20 | 49.13 |
| | HS2017 | $21.81 \pm 12.05^b$ | 0.50 | 55.83 |
| | HSC | $23.17 \pm 8.35^b$ | 4.15 | 43.80 |
| Pod weight per plant (PdWPl) | NS2017 | $13.39 \pm 4.39^a$ | 6.18 | 27.23 |
| | HS2016 | $10.08 \pm 3.96^b$ | 1.72 | 21.12 |
| | HS2017 | $10.51 \pm 6.15^b$ | 0.28 | 30.30 |
| | HSC | $10.15 \pm 3.70^b$ | 2.57 | 20.36 |
| Number of seed per pod (SdPd) | NS2017 | $3.26 \pm 0.51^a$ | 1.94 | 4.42 |
| | HS2016 | $1.93 \pm 0.52 \ c$ | 0.42 | 3.40 |
| | HS2017 | $2.38 \pm 0.73 \ b$ | 0.32 | 3.70 |
| | HSC | $2.04 \pm 0.49 \ c$ | 0.34 | 3.19 |
| Number of pods per plant (PdPl) | NS2017 | $8.22 \pm 2.05^b$ | 4.47 | 14.11 |
| | HS2016 | $12.67 \pm 4.65^a$ | 2.87 | 25.73 |
| | HS2017 | $9.22 \pm 4.72^b$ | 0.33 | 22.33 |
| | HSC | $11.76 \pm 4.07^a$ | 3.40 | 24.34 |
| 100 seed weight (100SdW) | NS2017 | $34.49 \pm 4.24^a$ | 22.49 | 44.67 |
| | HS2016 | $25.54 \pm 4.35^c$ | 11.87 | 35.84 |
| | HS2017 | $33.64 \pm 5.17^a$ | 14.48 | 44.56 |
| | HSC | $27.58 \pm 4.03^b$ | 14.83 | 35.93 |
| Pod harvest index (PHI) | NS2017 | $70.00 \pm 3.64^a$ | 60.96 | 78.12 |
| | HS2016 | $59.15 \pm 8.03^c$ | 26.58 | 74.38 |
| | HS2017 | $69.89 \pm 5.55^a$ | 50.00 | 83.99 |
| | HSC | $61.80 \pm 6.69^b$ | 37.79 | 76.29 |

(*Continued*)

**Table 1.** (Continued)

| Trait | Trial | Mean | Minimum | Maximum |
|---|---|---|---|---|
| Pods with well-formed seed (PdWF) | NS2017 | 69.52 ± 12.82[a] | 19.51 | 92.33 |
| | HS2016 | 31.18 ± 14.50[d] | 0.00 | 65.89 |
| | HS2017 | 56.29 ± 24.07[b] | 0.00 | 100.00 |
| | HSC | 37.37 ± 13.71[c] | 5.36 | 67.47 |
| Pods with shriveled seed (PdShr) | NS2017 | 3.02 ± 5.07[c] | 0.00 | 30.77 |
| | HS2016 | 29.38 ± 12.83[a] | 9.44 | 72.36 |
| | HS2017 | 12.67 ± 15.65[b] | 0.00 | 74.07 |
| | HSC | 25.22 ± 11.86[a] | 7.00 | 68.93 |
| Pods with partially well-formed seed (PdPWF) | NS2017 | 20.33 ± 10.08[a] | 6.37 | 56.81 |
| | HS2016 | 9.62 ± 5.46[b] | 0.00 | 32.57 |
| | HS2017 | 17.38 ± 12.10[a] | 0.00 | 100.00 |
| | HSC | 11.48 ± 5.40[b] | 2.05 | 41.28 |
| Pods with aborted embryos (PdAE) | NS2017 | 7.51 ± 5.28[c] | 0.00 | 31.66 |
| | HS2016 | 22.47 ± 9.79[a] | 1.75 | 59.93 |
| | HS2017 | 7.00 ± 8.34[c] | 0.00 | 40.00 |
| | HSC | 18.77 ± 8.06[b] | 3.94 | 56.21 |
| Pods that are empty (PdE) | NS2017 | 0.84 ± 1.37[b] | 0.00 | 7.87 |
| | HS2016 | 8.79 ± 11.52[a] | 0.00 | 78.88 |
| | HS2017 | 6.65 ± 13.42[a] | 0.00 | 88.00 |
| | HSC | 8.14 ± 10.06[a] | 0.00 | 81.96 |

HSC combines both heat stress trials.

Mean is shown with standard deviations.

Letters next to means indicate statistical differences according to Tukey test (p < 0.05).

scored as PdE was found significantly increased by factor ten and seven in both HS trials, respectively.

Taken together, heat stress affected various stages of reproductive development reducing productivity. In the HS2017 trial which experienced less extreme nighttime temperatures, some grain quality traits appeared similar to NS or in between HS2016 and NS values.

## Productivity in heat stress conditions is correlated to flowering time, pollen viability and seed formation

Evaluation of trait correlations between trials showed that all significant correlations that were identified were positive, indicating no extreme GXE effects (Table 2). Several significant correlations were found with the lower quality trial HS2017, and also correlations within trials were highly similar between HS2016 and HS2017 (S4 Fig), suggesting that the data are of sufficient quality to contribute to the analysis. DF and pollen traits were highly correlated between trials, also among stress and non-stress trials, suggesting that genetic variability is expressed stably across conditions (Table 2).

Also, for the yield component traits SdPd, 100SdW and PHI significant correlations were found between all trial combinations. In contrast, weak Yd correlations of low significance were only found between HS trials (0.25*). Similarly, yield components PdWPl, SdWPl, PdPl only resulted in significant correlations between HS trials, suggesting GXE effects between stress and non-stress conditions. Among pod defect classification traits, the only significant correlations were found in HS for traits PdShrSd and PdE. Taken together, generally more and

**Table 2. Phenotypic correlations between trials in heat stress (HS) and non-stress (NS).**

| Trait | Correlation coefficients | | | | | |
|---|---|---|---|---|---|---|
| | HS2016-HS2017 | | HS2016-NS | | HS2017-NS | |
| Days to flowering (DF) | 0.36 | *** | 0.49 | *** | 0.11 | ns |
| Pollen viability (PolVia) | 0.50 | *** | 0.56 | *** | 0.32 | ** |
| Indehiscent anthers (IA) | | | | | 0.29 | ** |
| Yield per ha (Yd) | 0.25 | * | 0.10 | ns | 0.13 | ns |
| Seed weight per plant (SdWPl) | 0.23 | * | 0.16 | ns | 0.09 | ns |
| Number of seed per plant (SdPl) | 0.22 | ns | 0.17 | ns | 0.11 | ns |
| Pod weight per plant (PdWPl) | 0.23 | * | 0.16 | ns | 0.08 | ns |
| Number of seed per pod (SdPd) | 0.32 | ** | 0.39 | *** | 0.27 | * |
| Number of pods per plant (PdPl) | 0.38 | *** | 0.19 | ns | 0.06 | ns |
| 100 seed weight (100SdW) | 0.44 | *** | 0.24 | * | 0.49 | *** |
| Pod harvest index (PHI) | 0.38 | *** | 0.34 | ** | 0.50 | *** |
| Pods with well-formed seed (PdWF) | 0.17 | ns | 0.09 | ns | 0.13 | ns |
| Pods with shriveled seed (PdShr) | 0.31 | ** | -0.03 | ns | 0.11 | ns |
| Pods with partially well-formed seed (PdPWF) | 0.16 | ns | 0.20 | ns | 0.13 | ns |
| Pods with aborted embryos (PdAE) | 0.08 | ns | 0.08 | ns | 0.16 | ns |
| Pods that are empty (PdE) | 0.36 | *** | 0.20 | ns | 0.07 | ns |

NS and HS refer to non-stress and heat stress trials.

Pearson correlations shown with significance levels *, **, *** as p values > = 0.05, 0.01 and 0.001, respectively.

higher correlations were found between HS trials, so expectedly HS trials are more similar to each other and different from NS trial.

Correlations between traits were evaluated in NS trial and a combined heat stress data set (HSC) (Fig 5). Highest correlations were observed between yield and yield component traits, in NS as well as HS conditions (Yd, SdWPl, SdPl, PdWPl, SdPd and PdPl). The only exception being the PdPl—SdPd pair, which showed a weakly negative correlation, an expected tradeoff as less seeds per pod may allow more pod formation. 100SdW is positively correlated to yield under both conditions and also with some yield component traits, moderately more pronounced under HS.

PHI showed distinctly higher trait correlations under HS than under NS. It was found positively correlated with yield (0.44), with most yield components and the proportion of non-defective pods (PdWF), but negatively correlated with traits that describe pod and seed formation defects. Also pollen traits had a strongly condition dependent response. Only under HS PolVia was correlated with yield and yield components (Fig 5). In contrast, IA shows predominantly the inverse pattern. Correlations between PolVia and IA was high at -0.81 and -0.51, respectively. In the same way, in HS DF was negatively correlated with yield and yield components. The only positive correlation is with seed formation defects such as PdShr and PdAE.

To further investigate the effect of flowering time in HS, we compared the retention of yield and flowering time under HS (Fig 6). %Yd in HS and %DF in HS are negatively correlated, i.e., those lines that extend their DF in HS eventually produce less. This indicates that heat stress is sensed earlier in development and the susceptibility response starts before flowering. PHI, early flowering and pollen traits are indicators for many desirable traits suggesting them as useful selection traits for performance under HS.

Among pod defect classification traits, the PdWF appeared positively correlated with yield traits and negatively linked to the pod defect traits. These traits, PdShr, PdAE and PdE, were in turn negatively correlated with yield traits in both conditions. Interestingly, pods evaluated

## Heat stress (HSC)

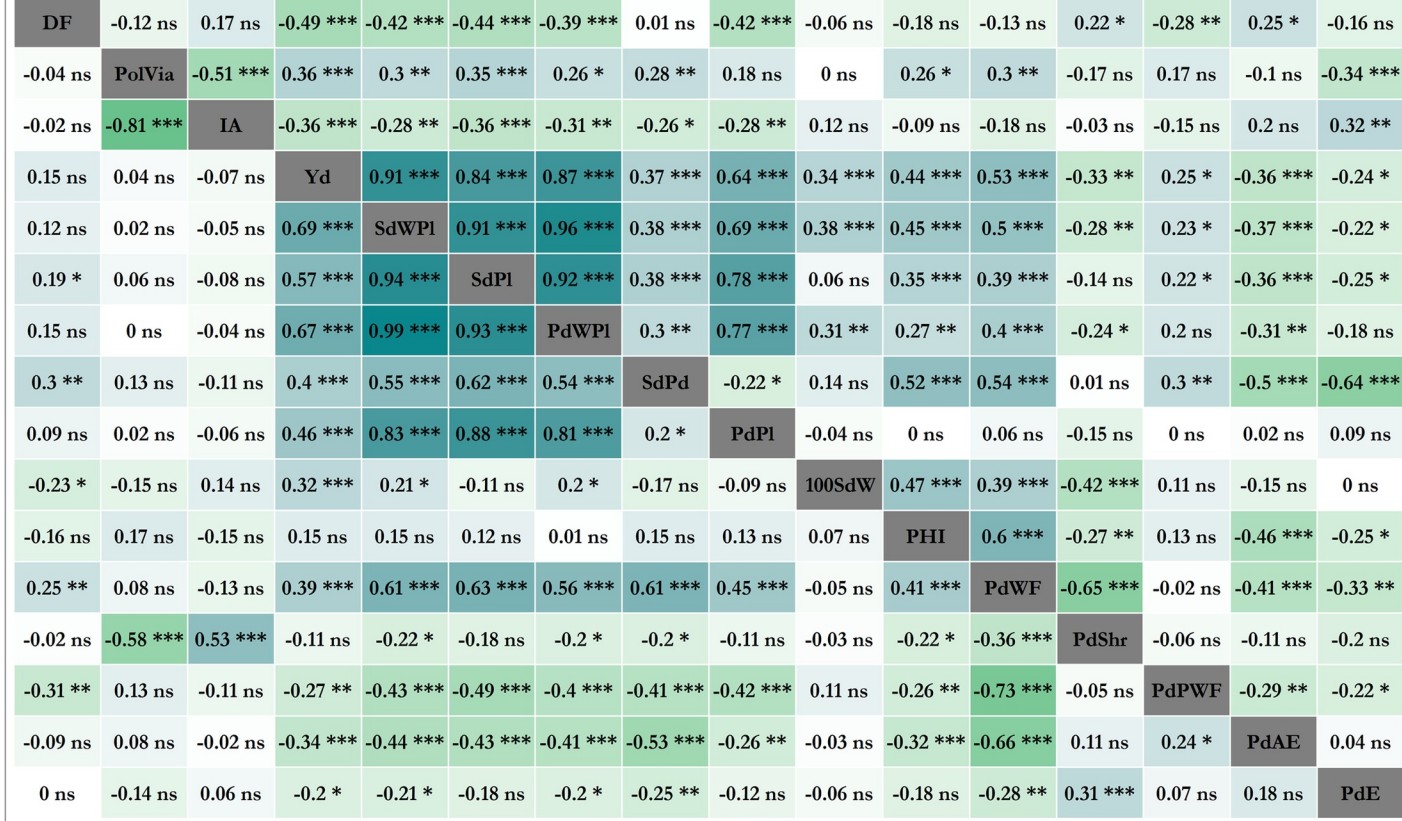

| | | | | | | | | | | | | | | | |
|---|---|---|---|---|---|---|---|---|---|---|---|---|---|---|---|
| **DF** | -0.12 ns | 0.17 ns | -0.49 *** | -0.42 *** | -0.44 *** | -0.39 *** | 0.01 ns | -0.42 *** | -0.06 ns | -0.18 ns | -0.13 ns | 0.22 * | -0.28 ** | 0.25 * | -0.16 ns |
| -0.04 ns | **PolVia** | -0.51 *** | 0.36 *** | 0.3 ** | 0.35 *** | 0.26 * | 0.28 ** | 0.18 ns | 0 ns | 0.26 * | 0.3 ** | -0.17 ns | 0.17 ns | -0.1 ns | -0.34 *** |
| -0.02 ns | -0.81 *** | **IA** | -0.36 *** | -0.28 ** | -0.36 *** | -0.31 ** | -0.26 * | -0.28 ** | 0.12 ns | -0.09 ns | -0.18 ns | -0.03 ns | -0.15 ns | 0.2 ns | 0.32 ** |
| 0.15 ns | 0.04 ns | -0.07 ns | **Yd** | 0.91 *** | 0.84 *** | 0.87 *** | 0.37 *** | 0.64 *** | 0.34 *** | 0.44 *** | 0.53 *** | -0.33 ** | 0.25 * | -0.36 *** | -0.24 * |
| 0.12 ns | 0.02 ns | -0.05 ns | 0.69 *** | **SdWPl** | 0.91 *** | 0.96 *** | 0.38 *** | 0.69 *** | 0.38 *** | 0.45 *** | 0.5 *** | -0.28 ** | 0.23 * | -0.37 *** | -0.22 * |
| 0.19 * | 0.06 ns | -0.08 ns | 0.57 *** | 0.94 *** | **SdPl** | 0.92 *** | 0.38 *** | 0.78 *** | 0.06 ns | 0.35 *** | 0.39 *** | -0.14 ns | 0.22 * | -0.36 *** | -0.25 * |
| 0.15 ns | 0 ns | -0.04 ns | 0.67 *** | 0.99 *** | 0.93 *** | **PdWPl** | 0.3 ** | 0.77 *** | 0.31 ** | 0.27 ** | 0.4 *** | -0.24 * | 0.2 ns | -0.31 ** | -0.18 ns |
| 0.3 ** | 0.13 ns | -0.11 ns | 0.4 *** | 0.55 *** | 0.62 *** | 0.54 *** | **SdPd** | -0.22 * | 0.14 ns | 0.52 *** | 0.54 *** | 0.01 ns | 0.3 ** | -0.5 *** | -0.64 *** |
| 0.09 ns | 0.02 ns | -0.06 ns | 0.46 *** | 0.83 *** | 0.88 *** | 0.81 *** | 0.2 * | **PdPl** | -0.04 ns | 0 ns | 0.06 ns | -0.15 ns | 0 ns | 0.02 ns | 0.09 ns |
| -0.23 * | -0.15 ns | 0.14 ns | 0.32 *** | 0.21 * | -0.11 ns | 0.2 * | -0.17 ns | -0.09 ns | **100SdW** | 0.47 *** | 0.39 *** | -0.42 *** | 0.11 ns | -0.15 ns | 0 ns |
| -0.16 ns | 0.17 ns | -0.15 ns | 0.15 ns | 0.15 ns | 0.12 ns | 0.01 ns | 0.15 ns | 0.13 ns | 0.07 ns | **PHI** | 0.6 *** | -0.27 ** | 0.13 ns | -0.46 *** | -0.25 * |
| 0.25 ** | 0.08 ns | -0.13 ns | 0.39 *** | 0.61 *** | 0.63 *** | 0.56 *** | 0.61 *** | 0.45 *** | -0.05 ns | 0.41 *** | **PdWF** | -0.65 *** | -0.02 ns | -0.41 *** | -0.33 ** |
| -0.02 ns | -0.58 *** | 0.53 *** | -0.11 ns | -0.22 * | -0.18 ns | -0.2 * | -0.2 * | -0.11 ns | -0.03 ns | -0.22 * | -0.36 *** | **PdShr** | -0.06 ns | -0.11 ns | -0.2 ns |
| -0.31 ** | 0.13 ns | -0.11 ns | -0.27 ** | -0.43 *** | -0.49 *** | -0.4 *** | -0.41 *** | -0.42 *** | 0.11 ns | -0.26 ** | -0.73 *** | -0.05 ns | **PdPWF** | -0.29 ** | -0.22 * |
| -0.09 ns | 0.08 ns | -0.02 ns | -0.34 *** | -0.44 *** | -0.43 *** | -0.41 *** | -0.53 *** | -0.26 ** | -0.03 ns | -0.32 *** | -0.66 *** | 0.11 ns | 0.24 * | **PdAE** | 0.04 ns |
| 0 ns | -0.14 ns | 0.06 ns | -0.2 * | -0.21 * | -0.18 ns | -0.2 * | -0.25 ** | -0.12 ns | -0.06 ns | -0.18 ns | -0.28 ** | 0.31 *** | 0.07 ns | 0.18 ns | **PdE** |

## Non-stress (NS)

**Fig 5. Phenotypic trait correlations in the IJR x AFR298 population, in non-stress (NS, lower left) and heat stress conditions (HSC, upper right).** For abbreviations see Table 1. Pearson correlations shown with significance levels *, **, *** as p values > = 0.05, 0.01 and 0.001, respectively.

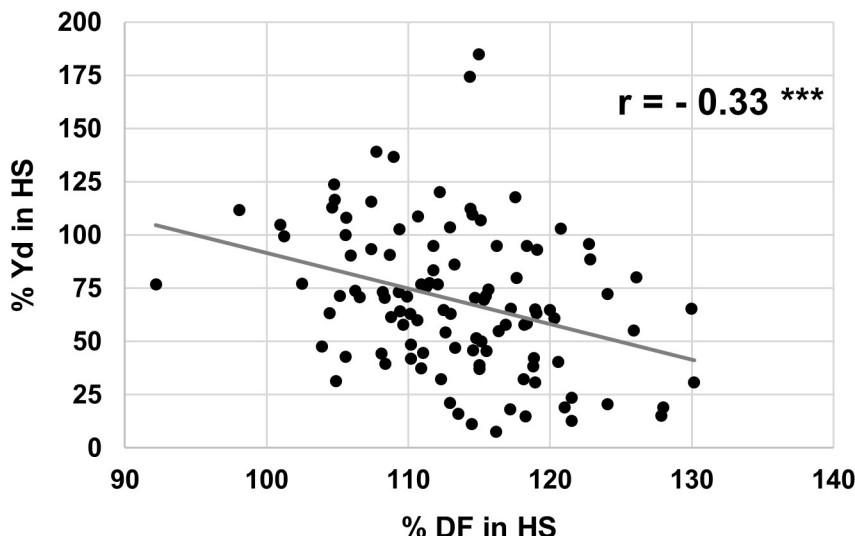

**Fig 6. Phenotypic correlation of % yield retention and flowering time under heat stress (in % compared to non-stress).** r represents Pearson's correlation.

with partial defects as PdPWF, was negatively associated with yield traits in NS, but positively in HS. In NS PdPWF appear to indicate problems, whereas in HS PdPWF is a sign of something still working. In summary, traits at different stages from floral induction, over pollen quality to seed fill are associated with good performance under HS. Productive plants under HS need to be early flowering, have viable pollen, and show good seed fill.

## QTL mapping of heat tolerance traits

The population IJR x AFR 298 was genotyped with the BARCBean6K_3 chip. Of the ~4800 SNPs with quality data (less than 50 missing data points) a fairly low number of 400 markers was polymorphic between the parents both of which belong to the Andean gene pool. After applying a binning filter, 162 informative markers remained to be used for QTL analysis. The genetic map covers, 1284 cM over 11 linkage groups, with an average marker to marker distance of 7.9 cM.

QTL mapping identified 19 QTLs for nine traits located on eight chromosomes (Fig 7, Table 3). The QTL PolVia5.1 was identified in both HS2016 and HSC data sets. The supporting allele originated from the more heat tolerant IJR parent, explaining 15 and 16% of the observed variance. For the most part, QTLs were detected in both HS2016 and the related HSC data set at the same time (9 out of 11 cases).

PHI5.1 was detected in close proximity to PolVia5.1 on chromosome Pv05 (16 cM) and the positive allele was in both cases contributed by IJR. PHI5.1 is one of three condition independent QTLs, that were detected in both HS and NS. PHI5.1 explains between 13 and 37% of the observed variance and also displayed the highest LOD significance of all QTLs. The traits are weakly positively correlated; hence, these QTLs may tag the same allelic variation associated with two traits.

A further PolVia QTL, PolVia8.1, was identified in the HS2017 trial, explaining a larger proportion of 20% of the variance, again with the IJR presenting the positive allele. At the same position, PdPl8.1 appeared in HS2016 representing two favorable traits associated with the IJR allele.

Next to PHI5.1, three further condition independent QLTs were found in both HS and NS conditions, DF1.1, SdPd1.1 and SdPd1.2. The QTL DF1.1 and SdPd1.1 fall within an interesting QTL hotspot. Allelic variation in this locus on chromosome Pv01 affects six different traits, three further yield component traits and seed formation defects as PdE, explaining between 10 and 31% of the variance. IJR allele leads to high seed numbers (SdPd1.1 and SdPl1.1), whereas the AFR298 allele is linked to embryo abortion, formation of more pods and large size of formed seed (PdE1.1, PdPl1.1, 100SdW1.1). SdPd1.2 was significant in all three data sets, explaining 11 to 16% of the variation with the positive allele based on the allele from AFR298.

## RIL lines with best genotypes and phenotypes

Lines with favorable alleles for tolerance to heat stress were identified, which can be a source for germplasm development. We evaluated the allelic effects of several major QTLs that were identified in multiple data sets. Markers for the QTLs PolVia5.1, PHI5.1, SdPd1.1, PdShr1.1, PdE1.1, and SdPd1.2 show a moderate effect on the overall observed variability, in line with a semi-quantitative inheritance (Fig 8).

No lines were identified that share all six QTLs with the favorable alleles for tolerance to heat stress. 11 RIL lines were found with the favorable alleles for the three QTLs PolVia5.1, PdShr1.1 and PdE1.1. The most promising four are shown in Table 4. RIS43 was among the top 25 yielding lines in all three trials, indicating yield stability also under NS. Also, RIS43 had good values for PolVia and low PdShr, next to the three QTLs. None of the top lines have good (low) scores for PdE, even though PdE is negatively related to yield. The disposition to form

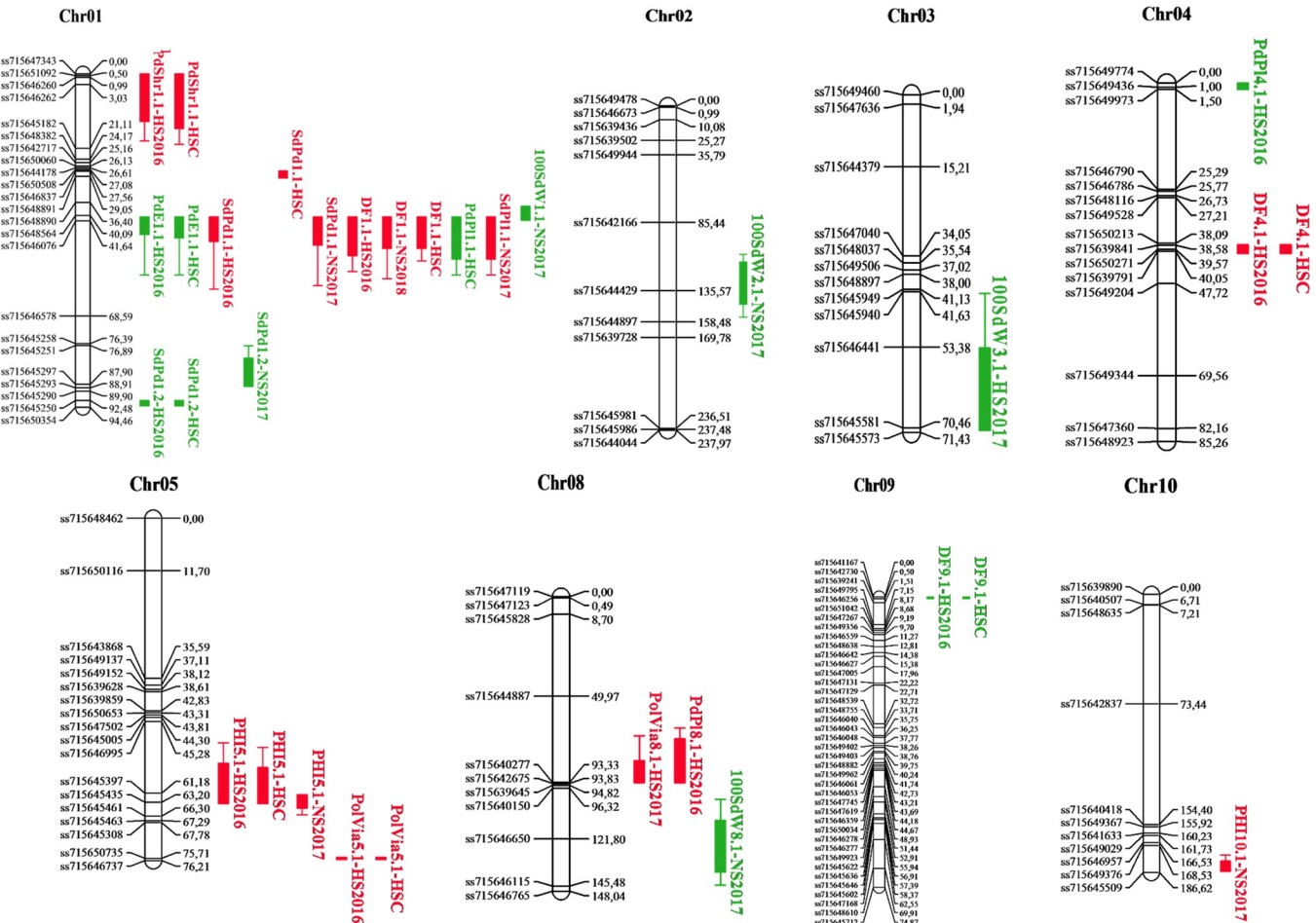

**Fig 7. QTLs identified in the IJR x AFR298 population evaluated under heat stress (HS) and non-stress (NS) conditions.** Evaluation of 107 lines with 162 polymorphic markers identified QTLs for nine traits. The eight chromosomes that harbor these QTLs are shown. The source of the positive allele is indicated in red for IJR and green for AFR298. For trait abbreviations see Table 1.

many pods may be more relevant for productivity, even though some of them are empty. These lines represent the best candidates for use in further germplasm improvement.

## Discussion

Climate change will negatively impact crop yields and with that food security. High temperatures are among the principal abiotic stresses limiting bean production [4]. In this work we investigated heat stress effects in a common bean population. Field trials were carried out in a region with higher temperatures where bean is not produced, whereas rice withstands conditions in this area. Such conditions of moderate heat stress resemble environmental conditions of future bean production areas in the tropics. They provide more relevant information than e.g., growth chamber experiments with extreme temperatures in artificial environments.

IJR x AFR298 population of the more heat sensitive Andean gene pool represents the preferred grain classes in Sub Saharan Africa and parts of south America [39]. A major focus of this work was to identify lines that produce quality grain in heat stressed field conditions. This work also presents first identification of QTLs under heat stress in common bean that may be used for molecular breeding tool development.

**Table 3. QTLs identified in the IJR x AFR298 population evaluated under heat stress (HS) and non-stress (NS) conditions.**

| QTL | Trial | Chr | Pos | CI | Left marker | Right marker | LOD | PVE% | Add | Source |
|---|---|---|---|---|---|---|---|---|---|---|
| **FLOWERING** | | | | | | | | | | |
| **Days to flowering** | | | | | | | | | | |
| DF1.1 | HS2016 | 1 | 42 | 40.5–51.5 | ss715646076 | ss715646578 | 3.07 | 10.42 | 0.75 | I |
| DF1.1 | NS2018 | 1 | 44 | 40.5–49.5 | ss715646076 | ss715646578 | 8.37 | 31.27 | 1.04 | I |
| DF1.1 | HSC | 1 | 42 | 40.5–49.5 | ss715646076 | ss715646578 | 3.01 | 10.80 | 0.74 | I |
| DF4.1 | HS2016 | 4 | 40 | 38.5–40.5 | ss715650271 | ss715639791 | 5 | 17.05 | 0.93 | I |
| DF4.1 | HSC | 4 | 40 | 38.5–40.5 | ss715650271 | ss715639791 | 4.16 | 14.85 | 0.84 | I |
| DF9.1 | HS2016 | 9 | 0 | 0.0–0.5 | ss715641167 | ss715642730 | 3.54 | 11.73 | -0.77 | A |
| DF9.1 | HSC | 9 | 0 | 0.0–0.5 | ss715641167 | ss715642730 | 3.79 | 13.39 | -0.79 | A |
| **Pollen viability** | | | | | | | | | | |
| PolVia5.1 | HS2016 | 5 | 76 | 75.5–76.0 | ss715650735 | ss715646737 | 3.68 | 15.1 | 5.21 | I |
| PolVia5.1 | HSC | 5 | 76 | 75.5–76.0 | ss715650735 | ss715646737 | 4.15 | 16.39 | 4.63 | I |
| PolVia8.1 | HS2017 | 8 | 93 | 82.5–93.5 | ss715644887 | ss715640277 | 5.1 | 20.12 | 5.64 | I |
| **HARVEST** | | | | | | | | | | |
| **Pods with shriveled seed** | | | | | | | | | | |
| PdShr1.1 | HS2016 | 1 | 3 | 0.0–13.5 | ss715646260 | ss715646262 | 2.84 | 11.42 | 4.33 | I |
| PdShr1.1 | HSC | 1 | 4 | 0.0–15.5 | ss715646262 | ss715645182 | 2.57 | 10.68 | 3.94 | I |
| **Pods that are empty** | | | | | | | | | | |
| PdE1.1 | HS2016 | 1 | 41 | 40.5–45.5 | ss715648564 | ss715646076 | 5.16 | 20.21 | -5.36 | A |
| PdE1.1 | HSC | 1 | 41 | 40.5–46.5 | ss715648564 | ss715646076 | 4.41 | 17.62 | -4.35 | A |
| **Pod harvest index** | | | | | | | | | | |
| PHI5.1 | HS2016 | 5 | 61 | 54.5–63.5 | ss715646995 | ss715645397 | 3.08 | 13.02 | 2.84 | I |
| PHI5.1 | NS2017 | 5 | 63 | 61.5–64.5 | ss715645397 | ss715645435 | 11.71 | 36.51 | 2.22 | I |
| PHI5.1 | HSC | 5 | 61 | 55.5–63.5 | ss715646995 | ss715645397 | 3.19 | 13.65 | 2.84 | I |
| PHI10.1 | NS2017 | 10 | 186 | 179.5–186.0 | ss715649376 | ss715645509 | 2.95 | 7.83 | 1.03 | I |
| **Number of seed per pod** | | | | | | | | | | |
| SdPd1.1 | HS2016 | 1 | 42 | 40.5–47.5 | ss715646076 | ss715646578 | 6.64 | 23.2 | 0.25 | I |
| SdPd1.1 | NS2017 | 1 | 41 | 40.5–48.5 | ss715648564 | ss715646076 | 5.66 | 19.21 | 0.23 | I |
| SdPd1.1 | HSC | 1 | 28 | 27.5–29.5 | ss715646837 | ss715648891 | 7.07 | 20.7 | 0.24 | I |
| SdPd1.2 | HS2016 | 1 | 94 | 92.5–94.0 | ss715645250 | ss715650354 | 3.52 | 11.41 | -0.17 | A |
| SdPd1.2 | NS2017 | 1 | 87 | 80.5–88.5 | ss715645251 | ss715645297 | 3.49 | 11.86 | -0.17 | A |
| SdPd1.2 | HSC | 1 | 94 | 92.5–94.0 | ss715645250 | ss715650354 | 5.50 | 15.61 | -0.2 | A |
| **100 seed weight** | | | | | | | | | | |
| 100SdW1.1 | NS2017 | 1 | 40 | 37.5–41.5 | ss715648890 | ss715648564 | 7 | 14.29 | -1.85 | A |
| 100SdW2.1 | NS2017 | 2 | 124 | 114.5–145.5 | ss715642166 | ss715644429 | 3.64 | 19.76 | -2.12 | A |
| 100SdW3.1 | HS2017 | 3 | 59 | 53.5–71.0 | ss715646441 | ss715645581 | 2.8 | 13.32 | -1.99 | A |
| 100SdW8.1 | NS2017 | 8 | 129 | 112.5–138.5 | ss715646650 | ss715646115 | 4.61 | 12.82 | -1.7 | A |
| **Number of pods per plant** | | | | | | | | | | |
| PdPl1.1 | HSC | 1 | 42 | 40.5–52.5 | ss715646076 | ss715646578 | 3.14 | 12.98 | -1.52 | A |
| PdPl4.1 | HS2016 | 4 | 0 | 0.0–1.5 | ss715649774 | ss715649436 | 3.74 | 8.01 | -1.71 | A |
| PdPl8.1 | HS2016 | 8 | 84 | 71.5–93.5 | ss715644887 | ss715640277 | 3.05 | 13.28 | 2.2 | I |
| **Number of seed per plant** | | | | | | | | | | |
| SdPl 1.1 | NS2017 | 1 | 42 | 40.5–52.5 | ss715646076 | ss715646578 | 2.85 | 11.63 | 3.04 | I |

Chr: chromosome, Pos: Genetic position in centimorgan, CI: Confidence interval in centimorgan, PVE: Phenotypic variation explained by the QTL, Add: Estimated additive effect of the marker, Sources: Parents AFR298 A or IJR I that contribute the positive allele. HSC combines both heat stress trials. For trait abbreviations see Table 1.

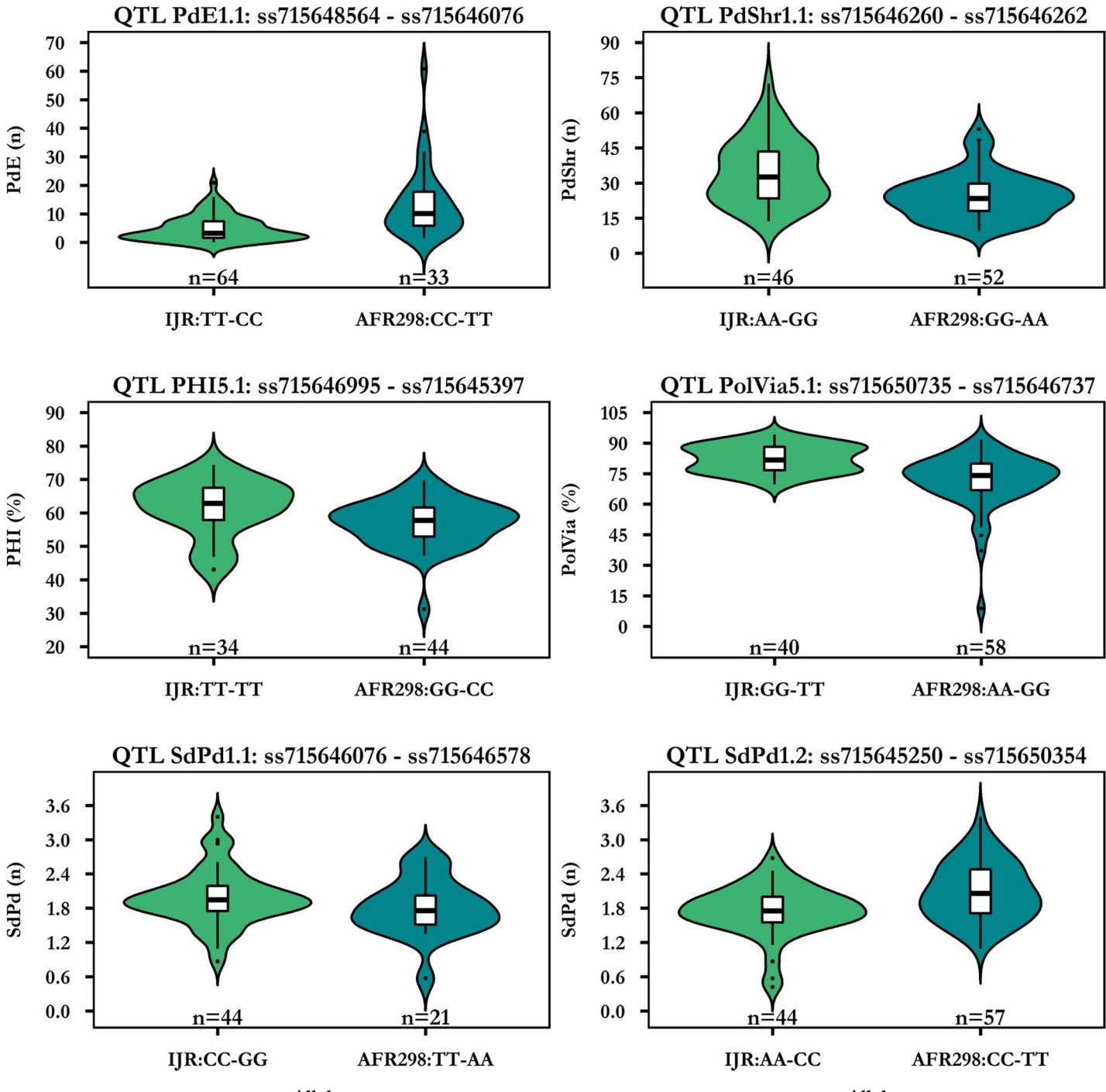

**Fig 8. Allelic effects of significant flanking marker pairs in the QTLs analysis.** Effects are shown as boxplots as well as violin plots for QTLs for the traits pods that are empty (PdE), pods with shriveled seed (PdShr), pod harvest index (PHI), pollen viability (PolVia) and seed per pod (SdPd) in the lines evaluated in HS2016. Only lines with homozygous marker calls are shown.

Evaluation of heat stress effects in field conditions has the inevitable weakness that stress and control trials cannot be conducted in the same location at the same time, as it is possible e.g., for drought stress by adding or withholding irrigation. Heat can only be controlled in very

**Table 4. Selected RIL lines that carry favorable alleles of QTLs for PolVia5.1, PdShr1.1 and PdE1.1 and their phenotypes.**

| Lines | Pollen viability HS2016 | Pods with shriveled seed HS2016 | Pods that are empty HS2016 | Yield HS2016 | Yield HSC | Yield NS2016 |
|---|---|---|---|---|---|---|
| RIS 43 | 78.06* | 14.27** | 7.55 | 1740*** | 1289** | 1966** |
| RIS 40 | 87.53** | 12.17*** | 8.3 | 1580*** | 1264* | 1359 |
| RIS 65 | 82.05* | 24.39* | 7.15 | 1306** | 1084* | 1675** |
| RIS 62 | 94.21*** | 14.58** | 6.62 | 1260* | 1171* | 1115 |

Asterisks indicate if the lines belong to the best 50% (*), best 25% (**) or best 10% (***) of the respective data set.

artificial environments such as greenhouses or growth chambers, which may not represent well farmer's production conditions. That means that the effects observed in the heat stress trials here could in part be location effects. In our heat stress trial the major affected traits were pollen viability, seed set and seed yield, which were also reported previously as heat stress effects under both controlled and field conditions [15, 17, 18, 21, 40, 41]; hence, the effects evaluated in this study are interpreted to be predominantly due to heat stress.

## Heat stress causes defects in multiple stages of reproductive development

Studies on heat stress in common bean have identified several defects during the reproductive phase [15, 42]. Studies have indicated that high nighttime temperatures negatively affect reproductive development more than maximal day temperatures [41]. In this work where two heat stress trials were conducted, stronger stress effects were observed in H2016 which had higher nighttime temperatures, compared to HS2017 which had higher daytime temperatures. These results provide further evidence to the major importance of nighttime temperatures in heat stress, in line with previous reports [16].

Flowering time was delayed in HS. Wallace et al [43] reported delayed flowering in HS in bean but associated with photoperiod sensitivity in genotypes, whereas early flowering and maturity was reported as a result of HS in groundnut [44], in *Cichorium intybus* [45] or in lentil [46]. While this observation may be influenced by the location, we showed that in heat stress delay in flowering is correlated with yield reduction, hence, these effects appear to be linked.

High pollen viability appeared to be an indicator of heat tolerance. Non-viable pollen and failure of anther dehiscence were reported to be a major defect caused by high temperature treatment [17, 18], also observed in other crops like lentil [46]. Defects in embryo development and seed set were observed leading to partially empty or empty pods. Defects in seed set were previously reported [17], as well as empty pods [47], that were considered parthenocarpic [17]. Subsequent seed fill of developing seed was also compromised leading to shriveled and undersized seed, also in line with previous reports in bean [40] or in soybean [48]. A higher number of pods per plant was observed in heat stress conditions, in line with an extended vegetative phase as a result of failing seed set. This indicates that defects in early reproductive phases may be compensated to some extent by increasing the number of flowers and embryos. All these factors lead to a reduction of final yield and yield components. Yield reductions as a result of heat have previously been reported from greenhouse [18, 41, 49] as well as in field [18, 50] trials.

Heat stress causes defects along all phases of reproductive development, from the onset of flowering to seed fill. This indicates a general process of heat stress perception that affects all reproductive tissues. Reduced concentration of free hexoses were suggested as a mechanism, and reduced hexose levels were found in heat stressed bean together with corresponding transcriptome changes [51]. Limitations in phloem loading at source tissues may consequently starve the developing flowers, developing pollen and newly fertilized seeds. This is consistent

with results in rice where drought and heat resulted in sugar starvation in reproductive organs [52] and observation from maize, where heat did not affect the photosynthetic rate, but resulted in lower sugar transport to reproductive tissues [53]. Even though bean germplasm candidates were identified that show tolerance in specific processes such as pollen viability or seed fill, there appears to be an overarching stress tolerance mechanism independent of specific processes that leads to tolerant germplasm.

## Breeding for heat tolerance: Physiological indicators and molecular tools

Climate change models predict that large regions in Africa will become unsuitable for bean production by 2050 unless heat adaptation is improved [13, 14]. On the other hand, increasing heat tolerance by 2.5˚C is expected to benefit 7.2 million hectares, allowing to increase production areas including currently unsuitable areas [4]. Germplasm with improved tolerance needs to be developed quickly, to provide farmers with the right materials for now and for the future.

PolVia has been proposed as an indicators for heat tolerance that could be employed in breeding selection [17]. Evaluation of anthers for indehiscence requires more skill but is eventually easier and faster than pollen evaluations, requires no extraction, coloring, image capture and analysis. However, the application developed for image analysis of pollen viability used in this study made this process much more efficient than previously. QLTs were only found for PolVia and PolVia had slightly higher trait correlations, so this trait may be more valuable.

Pod harvest index (PHI) was shown to be an indicator for yield in several stress conditions [54] and it also appears to be useful for improving drought tolerance. Seed fill is also important for marketability as shriveled seed of reduced size have very low market value. Hence, seed quality traits need to be a primary focus in breeding, next to yield. The traits PolVia and PHI appear useful for selection for heat tolerance; however, they are cumbersome and resource demanding to phenotype, and phenotypic evaluation may not be informative in all conditions. For this reason, a molecular marker would be very useful.

González et al [55] reported QTLs for several traits, such as number of seed, seed per pod, pods per plant and seed weight in a similar region of chromosome Pv01 as the QTL hotspot for seed traits described in this work. A phenology QTL on chromosome Pv01 had been reported in several studies and candidate genes were described by [56]. A QTL hotspot affecting numerous traits may be difficult to employ in breeding due to the pleiotropic effects. QTL PHI5.1 has been previously identified in a similar position [57, 58], and presents a good target for MAS as PHI is associated to yield under many conditions. QTLs identified here should be confirmed in more complex genetic backgrounds to validate their usefulness in breeding.

Well performing lines have been identified in this work, that carry several good alleles and have generally good performance. Breeding for heat tolerance has been successful in other crops such as maize, employing selection in stress prone target regions [59]. Indirect selection through secondary traits with high heritability and associated with grain yield under stress have been reported to be an effective approach in stress tolerance breeding compared with direct selection for grain yield [60].

In this and other studies a good genetic variability for heat tolerance was observed, hence, a combination of phenotypic selection for yield under HS and indicator traits related to HS, together with MAS seems to be a promising strategy for development of tolerant germplasm.

## Outlook

Climate change will have an increasing effect on bean productivity in many areas of the world and a better understanding of heat stress will help to prepare for the future. We confirmed the importance of sensitive processes such as pollen shed, seed set and seed fill, and observed that

grain quality needs to be considered in breeding. In addition, heat stress effects spanning the whole reproductive phase from flower induction to harvest suggest an overarching mechanism of heat susceptibility that is independent of specific processes. More efficient breeding methods are required, employing physiological indicators of heat tolerance and molecular tools to generate the germplasm that farmers need to face the future.

## Supporting information

**S1 Fig. Minimum and maximum daily temperatures for the Palmira location (NS2017) between the months of July and October of 2017 and for the Alvarado location between the months of July and October of 2016 and 2017 (HS2016 and HS2017).**
(TIF)

**S2 Fig. Viable and non-viable pollen grain stained with acetocarmine dye evaluated on a microscope slide.**
(TIF)

**S3 Fig. Phenotypic distributions of evaluated traits under heat stress (HS) and non-stress (NS) conditions.** Parental lines IJR indicated as blue dot and AFR298 in red.
(TIF)

**S4 Fig. Phenotypic correlations within the heat stress (HS) trials in 2016 and 2017 in the IJRxAFR298 population.**
(TIF)

**S1 Table. Complete list of features extracted by image analysis from pollen images.** The machine learning algorithm to classify viable and non-viable pollen was based on these features.
(XLSX)

## Acknowledgments

We thank Dr Qijian Song at ARS Beltsville for support with the genotyping. We thank the CIAT bean team for their great support.

## Author Contributions

**Conceptualization:** Bodo Raatz.

**Formal analysis:** Yulieth Vargas, Bodo Raatz.

**Investigation:** Yulieth Vargas, Victor Manuel Mayor-Duran, Hector Fabio Buendia.

**Methodology:** Henry Ruiz-Guzman.

**Project administration:** Bodo Raatz.

**Software:** Henry Ruiz-Guzman.

**Supervision:** Victor Manuel Mayor-Duran, Bodo Raatz.

**Visualization:** Henry Ruiz-Guzman.

**Writing – original draft:** Yulieth Vargas, Bodo Raatz.

**Writing – review & editing:** Bodo Raatz.

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
