## [Decision Letter · Decision Letter 0]

18 Nov 2020

PONE-D-20-34193

Physiological and genetic characterization of heat stress effects in a common bean RIL population

PLOS ONE

Dear Dr. Raatz,

Thank you for submitting your manuscript to PLOS ONE. After careful consideration, we feel that it has merit but does not fully meet PLOS ONE’s publication criteria as it currently stands. Therefore, we invite you to submit a revised version of the manuscript that addresses the points raised during the review process.

We look forward to receiving your revised manuscript.

Kind regards,

Roberto Papa, PhD

Academic Editor

PLOS ONE

Journal Requirements:

2. Please ensure that you refer to Figure 4 in your text as, if accepted, production will need this reference to link the reader to the figure.

Reviewers' comments:

Reviewer's Responses to Questions

**Comments to the Author**

1. Is the manuscript technically sound, and do the data support the conclusions?

Reviewer #1: Partly

Reviewer #2: Yes

2. Has the statistical analysis been performed appropriately and rigorously? 

Reviewer #1: N/A

Reviewer #2: Yes

3. Have the authors made all data underlying the findings in their manuscript fully available?

Reviewer #1: Yes

Reviewer #2: Yes

4. Is the manuscript presented in an intelligible fashion and written in standard English?

Reviewer #1: No

Reviewer #2: Yes

5. Review Comments to the Author

Reviewer #1: First, I would like to highlight the importance and the need for studies involving high-temperature stress, mainly with the provisions of the increase in global average temperature in the coming years. As is well known, common beans are highly sensitive to extreme temperatures. In this sense, I would like to congratulate all authors for their initiative in working with such a complex trait, and to CIAT for their commitment to the genetic breeding of bean crop.

Due to the importance and the difficulty of working with high-temperature, I recommend the current manuscript for publication after "major revision" of the content, English review, and the correct formatting for the style of the journal.

General considerations:

I do not have the ability to evaluate the English, but the manuscript has many formatting errors, lack of standard and many repeated keywords, especially at the beginning of the paragraphs. A complete language review may be required. Throughout the text, numbers less than 10 did not follow a pattern, sometimes appear in full (“three buds” L161) and sometimes not (“5 plants” - L168). Also lack of standard terminology, such as the use of the word "variable" and "trait" for the same meaning throughout the text.

Another important point is the lack of the main objective of the manuscript. For a study of “Physiological characterization of heat stress”, the best would not be a biparental population, where the entire source of variability is restricted to both parents used, and the number of environments tested was very low (complete experiments only HS2016 and NS2017). At the same time, for a study of “Genetic characterization of heat stress”, the study is incomplete. The methodology for constructing the genetic map of the population is not even mentioned and the association model used (interval mapping) is outdated with the development of composite interval mapping (CIM). Currently, multiple interval mapping (MIM) is the model with the greatest statistical power and has been the most used for genetic mapping studies. Also, a large part of the text describes the development of software HYRBEAN for counting pollen grains which does not fit the objectives mentioned by the authors and is not discussed in the discussion session. In my view, there is no problem that the study has several objectives, however, there must be a harmony between them, and in the case of all being of equal importance, that all is treated with the same relevance and rigor.

Regarding the experiments, the authors describe the HS2016, HS2017, NS2017, and NS2018 trials, however, only the complete tests (with repetitions and with experimental design) HS2016 and NS2017 are valid and should be used. The combined analysis (HSC) does not bring any advantage, since it inserts even more environmental variation when adding the HS2017 (without repetitions and design) in the HS2016 trial. Another critical point is that the complete experiments conducted to the environment with high temperature and not high temperature (HS2016 and HS2017), were carried out in different years so that in general HS2016 presented maximum temperatures below than HS2017 (Supplementary Figure 2).

My specific considerations are in the attached file.

Reviewer #2: On the whole the paper is well written and the work carried out flawlessly.

the few things I would like to point out are:

row 30 (Abstract) - it is not specified respect to what there was a decrease of 37% and 26% in the 2016 and 2017 seasons.

row 90-91 (Introduction) - in these lines the citations have been indicated in a different form than the rest of the paper.

row 236 (Data Analysis) - I suggest starting the paragraph with "Phenotypic data analysis".

row 237 (Data Analysis) - " HS2016 y NS2017" maybe "y" is a typing error or a Spanish residue.

row 319 - 327 (Result) - refers to GXE effects and after GxE ... maybe a typing error.

tabel 1-3 -there is a reference to NS2018, maybe is a typing error.

Furthermore, I suggest more details regarding the genetic mapping and identification of QTL,

the paper does not specify which "statistical strategy" is used through the QTL IciMapping software,

(for example the use of the "Inclusive composite interval mapping" method can be specified, or even the threshold

of statistical significance in QTL analysis).

Finally, I suggest to include in the discussions a reflection on the fact that this trial was carried out in the open

field and in two different locations, and that this may affect the relationship between a phenotype and its association

with heat stress conditions affecting the QTL mapping interpretation. This is because the different environmental conditions between the two locations were not considered in the paper except for the temperature. Perhaps specifying why this test was performed in the field

and not in the greenhouse and consequently the relative advantages and disadvantages.

6. PLOS authors have the option to publish the peer review history of their article (what does this mean?). If published, this will include your full peer review and any attached files.

Reviewer #1: **Yes: **Caléo Panhoca Almeida

Reviewer #2: No

---

## [Author Response · Author response to Decision Letter 0]

24 Feb 2021

Response to reviewers

Reviewer #1: 

First, I would like to highlight the importance and the need for studies involving high-temperature stress, mainly with the provisions of the increase in global average temperature in the coming years. As is well known, common beans are highly sensitive to extreme temperatures. In this sense, I would like to congratulate all authors for their initiative in working with such a complex trait, and to CIAT for their commitment to the genetic breeding of bean crop.

Due to the importance and the difficulty of working with high-temperature, I recommend the current manuscript for publication after "major revision" of the content, English review, and the correct formatting for the style of the journal.

RESPONSE: We thank reviewer 1 for the comments and suggestions. He/she made and extensive effort to evaluate several aspects of the paper that will help to improve the quality of the manuscript, for which we are very grateful. 

General considerations:

I do not have the ability to evaluate the English, but the manuscript has many formatting errors, lack of standard and many repeated keywords, especially at the beginning of the paragraphs. A complete language review may be required. Throughout the text, numbers less than 10 did not follow a pattern, sometimes appear in full (“three buds” L161) and sometimes not (“5 plants” - L168). Also lack of standard terminology, such as the use of the word "variable" and "trait" for the same meaning throughout the text.

RESPONSE: We reviewed the document to improve the text according to the suggestions. The language was revise, as well as the number formatting in the text and the reference formatting style. 

Regarding variable and trait: I have seen several software packages that manage breeding programs and also trait ontology projects that use the term variable for what is usually named a trait. So that is quite common. But we are happy to change the text to use the term trait throughout. 

Another important point is the lack of the main objective of the manuscript. For a study of “Physiological characterization of heat stress”, the best would not be a biparental population, where the entire source of variability is restricted to both parents used, and the number of environments tested was very low (complete experiments only HS2016 and NS2017). At the same time, for a study of “Genetic characterization of heat stress”, the study is incomplete. The methodology for constructing the genetic map of the population is not even mentioned and the association model used (interval mapping) is outdated with the development of composite interval mapping (CIM). Currently, multiple interval mapping (MIM) is the model with the greatest statistical power and has been the most used for genetic mapping studies. Also, a large part of the text describes the development of software HYRBEAN for counting pollen grains which does not fit the objectives mentioned by the authors and is not discussed in the discussion session. In my view, there is no problem that the study has several objectives, however, there must be a harmony between them, and in the case of all being of equal importance, that all is treated with the same relevance and rigor.

RESPONSE:

The objective of this project was to perform a physiological and genetic evaluation of heat stress effects in the AFRxIJR RIL population. We performed the analysis of several physiological aspects and performed a genetic analysis, so we consider these terms to be applicable. We do not claim this to be an exhaustive study of these topics of course, which would be beyond the scope of this or any manuscript. Naturally, one could obtain more information using a larger population with more genetic variation, but for the available resources a bi-parental RIL population offers unmatched genetic resolution. 

Thank you for pointing out that the description of the genetic analysis was insufficient. We improved this part in the M&M section. 

The HYRBEAN tool is not a major objective in this study, but it was employed and found to be very useful. It hasn’t been published and the concept is quite novel, so we had to dedicate some space in the M&M section to describe it. We shortened the description a little and it takes up less than 5% of the document. 

Regarding the experiments, the authors describe the HS2016, HS2017, NS2017, and NS2018 trials, however, only the complete tests (with repetitions and with experimental design) HS2016 and NS2017 are valid and should be used. The combined analysis (HSC) does not bring any advantage, since it inserts even more environmental variation when adding the HS2017 (without repetitions and design) in the HS2016 trial. Another critical point is that the complete experiments conducted to the environment with high temperature and not high temperature (HS2016 and HS2017), were carried out in different years so that in general HS2016 presented maximum temperatures below than HS2017 (Supplementary Figure 2).

RESPONSE:

The quality of trials are best judged based on their informative value, rather than on the number of reps. The HS2017 trial is important to support the general findings from HS2016 on yield reduction, pollen viability, PHI and others. Without this support one could not say if these are only season specific observations, which would weaken conclusions in the manuscript. We added suppl Fig 4 that shows that trait correlations within HS2016 and HS2017 trials are highly similar which demonstrates that the increased noise in the lower quality trial is not masking the genetic effects. A trial of such poor quality that would render it useless for analysis would not show highly significant correlations. The same counts for the DF data set from NS2018. An absence of high correlations may be meaningless, but the observation of highly significant correlations for the highly heritable DF trait indicates a good informative value. 

The combined heat stress analysis HSC is the best available data set for heat stress. Fig 5 and suppl fig 4 do not indicate that the combination of all available data is inferior to any single trial. 

Yes, the temperature profiles of HS2016 and HS2017 were not identical, which is somewhat expected in open field experiments. This gives insights into the importance of nighttime temperatures, over daytime temperatures. To see the same general effects of heat in slightly different conditions, strengthens the notion that these are indeed heat effects. HS2017-NS correlations were higher for 100SdW and PHI, compared to HS2016-NS, suggesting that HS2017 is a less heat affected trial providing good quality data. 

My specific considerations are in the attached file.

 Specific considerations: 

The “Short Title” should not be equal to “Full Title". 

RESPONSE: I am not aware of such a rule, but I will be happy to follow the editor’s suggestions here. The short title is limited by number of characters. If the title is already short enough, I see no need to change it to make a new short title. 

Abstract 

The Abstract should not exceed 300 words and cannot be divided into paragraphs. 

RESPONSE: Thank you for noting that. We revised the abstract. I find no mentioning of paragraphs in the author guide, we will be happy to follow editorial suggestions. 

L23 – Change “Phaseolus vulgaris” by “Phaseolus vulgaris L.”. 

done

L27 - Change “RIL” by “RIL (recombinant inbred lines)”. 

done

L27 – Change “genepool” by “gene pool”. 

RESPONSE: We are also happy to use “gene pool”, both versions are used in the literature. 

L37 – Change “QTL” by “QTL (quantitative trait loci)”. 

done

L37 – Change “chromosome 5” by “chromosome Pv05”. Please do this for the entire manuscript. 

RESPONSE: changed. This is a bit of a relict from the early times of mapping and several publications don’t use that any more. Does the reviewer see a major advantage in the use of that term? 

L38 – The phrase “with the positive allele contributed by the more heat tolerant parent IJR” sounds strange. 

modified

L40 – Change “A QTL hotspot on chromosome 1 affected 6 phenology and seed formation traits” by “A QTL hotspot on chromosome Pv01 was mapped associated with six phenology and seed formation traits. 

Changed (and subsequently shortened)

Introduction 

The introduction covers the main points of the study, however, the lack of zeal in the accuracy of the references cited and in the formatting of the file is evident. The subject of QTL mapping was presented in a confusing and poorly introduced way, missing the link between the paragraphs. 

According to the journal style, citations must be between “[]” and not “()”. Please review the entire file. 

RESPONSE: We reviewed the section and modified the introduction of QTL mapping. Reference style was corrected.

L51 – The reference “(5)” does not seem to support the affirmation, so it is not necessary. 

Removed

L52 – Add “the” before “effects”. 

added

L52 – The phrase “However, climatic conditions are shifting due to effects of climate change” sounds strange. 

Modified and shortened

L56 – The phrase “Current research suggests that the average global temperatures have increased by ~0.8°C since 1880” need to be referenced. 

added

L60 – Change “as” by “than”. 

changed

L63 – The introduction paragraphs must not be separated with empty lines. Please review the entire file (L72, L77, L83, L94). 

reviewed

L65 – The phrase “High nighttime temperatures during the reproductive phase cause heat stress in common bean, 66 and to a lesser degree, high daytime temperatures” need to be referenced. 

referenced

L67 – Add “the” before “abortion”. 

added

L70 – Add “the” before “night”. 

added

L73/76 – I did not find any results in the reference “(16)” that supports the paragraph “Common bean genotypes of the Andean gene pool are commonly grown at mid to mid-high altitudes (1400- 2800 masl) or in cooler climates, whereas genotypes of the Mesoamerican gene pool adapt to low to mid altitude ranges (400 - 2000 masl) with higher temperatures. For this reason, Andean beans are expected to be more sensitive to high temperatures (16).” 

RESPONSE: That was indeed the wrong citation. Corrected to reference (4)

L79 – Add references of studies carried out “under stress conditions” and “controlled environments”, separately. 

RESPONSE: Some papers describe climate chamber, greenhouse as well field experiments, so this separation can not be made. 

L82 – Change “(Blair, Hoyos, Cajiao, & Kornegay, 2007)” by the journal style. 

changed

L84/88 – The paragraph is very confused and poorly structured. The QTL mapping makes it possible to identify loci associated with the trait of interest, in order to provide information on markers linked to the QTL that can be used for SAM. However, it is necessary to clarify that SAM and mapping are two totally different approaches. References are also missing. 

RESPONSE: Changed and shortened

L86 – Change “marker assisted” by “marker-assisted”. 

changed

L88 – Change “are” by “is”. 

changed

L90 – Change the reference by the journal style. 

changed

L92/93 – There is no point in discussing MAS if the work did not aim at MAS. It is preferable to improve the discussion of the importance of identifying markers associated with QTLs, aiming at the use of SAM. 

Text revised and shortened

L96 – Remove “(Indeterminate Jamaica Red)”. 

removed

L100 – Change “germplasm” by “lines”. 

Changed

Materials and methods 

Although a population with 107 genotypes is considered small for linkage mapping studies, it is necessary to consider the complexity of assessments for high-temperature stress. I have no experience with alpha lattice design and therefore I am not able to make considerations. However, in my opinion the NS2018 and HS2017 trials should not be considered for the study since both have no design and repetition. 

RESPONSE: As explained above both trials NS2018 and HS2017 show relevant correlations to the replicated trials and similar correlations within trials and thereby show that the data quality is good enough to add to the analysis. They contribute valuable insights to the discussed topics so we see value in their inclusion. 

L103 – Remove “(recombinant inbred lines)”. 

reorganized

L105 – Remove “(abbreviated from Indeterminate Jamaica Red)”. 

modified

L108 – “with wide adaptability” for what? Add the reference. 

explained

L114 – Remove “International Center of tropical agriculture”. The abbreviation CIAT has already been mentioned. 

correct

L119 – Remove “sowing at”. 

removed

L121/128 – In my opinion, that paragraph should be moved to the results section. 

RESPONSE: We agree and removed this text here. 

L122/128 – It is necessary to make it clearer that the variation of the maximum temperature mentioned, refers to the variations of the maximum temperatures of each day in relation to the total period of evaluation. The same for the minimum temperature variation. 

RESPONSE: The relevant differences are pointed out in results and discussion sections. 

L131,138, 145, 168 - All paragraphs begin the same "For evaluations". Rewrite. 

RESPONSE: We changed a few. Wording of the methods section only aims for clarity, we don’t see a problem with using repetitive, similar wording to describe similar activities. 

L134 – What means “experimental plots were not replicated”? For HS2017 the 3 repetitions mentioned for HS2016 were not adopted? I did not understand. 

RESPONSE: This is correct, the trial was not replicated. We modified the text to attempt to make that clearer. 

L143 – Was the trial irrigated? If so, how many times a day? Likewise for all trials? 

RESPONSE: We added the irrigation. Trials are usually watered when needed, at the most once per week. 

L145 – Add “period” or “time” after “flowering”. 

added

L145 and L168 - In my opinion it would be better if the authors use the phrases as sub-sections (level 3). (e.g. Evaluations during flowering period / Evaluations during harvest time) 

reformatted

L146 - In field evaluations, where the germination of the seeds presents great variation, the most recommended is the number of days for flowering to be given by the difference in the number of days of germination and flowering. 

RESPONSE: I do not recall any great variation in germination. The trait described here is the standard method, used in many other publications. 

L149 – Change “suggested” by “proposed” and correct the reference for the journal style. 

changed

L148/158 – The methodology description is extensive. The authors could cite the protocol used (Polanía et al 2016) and only highlight possible changes in the methodology. 

RESPONSE: We shortened the text a bit. Information is provided in the section that goes beyond Polania et al (2016). We do feel that we need to describe the method well, to allow the reader to easily understand the explanation of the novel HYRBEAN tool.

L160 – Change the phrase by “Finally, to determine the percent viability (number of grains stained with respect to the total), the pollen grains were count using the software HYRBEAN”.

changed 

L162 – Change “read” by “evaluated”. 

replaced

L166 – The caption must be more complete, adding the name of the species or "common bean", the name of the genotype used for the example and etc. 

RESPONSE: Caption was modified. This is not genotype specific.

L169 – Change “variables” by “trait”, 

Changed

L170 – Separate each trait on a line. 

changed

L171, 173, 174 – Change “formed” by “harvested”. 

modified

L175 – Remove “providing information on grain size.” 

removed

L195/234 – Regarding this sub-section, I can see two possible options: 

1: The development of the HYRBEAN software is rewritten more succinctly and the accuracy (correlation) is just mentioned. 

2: The development of the HYRBEAN software enters the objectives of the study, and the results (L229 / 234 and Fig 3) pass to the results section and are discussed later in the discussion session. 

RESPONSE: The software is a tool that was developed and used in this project, but not a major objective of this publication. We shortened the section a little. The accuracy is just mentioned in the last sentence, where it is necessary to demonstrate the tools functionality. It doesn’t represent data of this population so it should not be presented in the results section. 

L237 – Change “y” by “and”. 

changed

L244 – The package “ggplot2” does not perform statistical calculations. 

correct

L252 – Were the F6 lines used for extraction? If yes, why was the DNA pooled? 

Information added

L255 – Change “6000” by “5,398”. 

Thanks for the correction

L254 – Change “(1992)” by the journal style. 

changed

L259 – Remove “completely related or”. 

removed

L258/264 – The number of SNP filtered both for the polymorph of the parents and for the redundancy test can be presented in the results section. 

RESPONSE: both values 400 and 162 are mentioned.

Results 

L267 – Change “Heat stress (HS)” by “HS”. 

L268 – Change “non-stressed (NS)” by “NS”. 

RESPONSE: Given that many readers start reading the results section, we prefer to leave this information here for clarity. 

L269 – (Fig ??). 

4. It’s shown correctly in the word file for some reason, let’s see what the pdf makes out of it.

L271 – “During the reproductive phase the highest minimum daily temperatures (nighttime temperatures) were also registered in HS2016, whereas greatest maximum temperatures (daytime) were observed in HS2017.” is confused. 

Sentence modified

L274 – The graph used is very good! It clearly shows that there was a contrast between both environments (HS and NS). 

279 – Change “or” by “and”. 

changed

L280/281 – One more reason not to consider HS2017 data. 

RESPONSE: Out of any two trials, one will have a higher variation than the other. That doesn’t mean that the data is not useful. 

The manuscript needs a review of English and formatting for the style of the journal, so I do not comment further on these errors. 

L282 – “Onset of flowering was noted about 5 days later in the HS trials than in NS conditions.” Was the germination day of the plots recorded? Germination in the field depends intrinsically on soil moisture. Please provide more information about this if the authors want to discuss the difference in the number of days for flowering. 

RESPONSE: Germination was not quantified. No unusual gemination was observed and there is no reason to believe in a strong environmental effect on germination here. Seed are always sown in adequate soil conditions, usually after a preceding irrigation treatment. 

L319/320 – The affirmation is not valid. Although most correlations are significant (4 non-significant), the highest significant correlation was 0.5 and the lowest was 0.23 (mean 0.34). These values do not support that the data are of sufficient quality to contribute to analysis. 

RESPONSE: As mentioned above, significant positive correlations throughout the data sets show that the same genetic effects are observed in both trials. A data set that is very noisy and of so low quality to render it unsuitable for analysis would not show these clear correlations. It would be great if the reviewer could explain how else he or she interpreted these results and where he/she sees data that suggest that it is better not to use this data set. Just because a trial is not replicated does not mean that it is useless. I have seen many trials that were replicated and they were still useless. The similarity to related trials is the important factor to judge if a data set is valuable, not the count of the number of replicates. The 2017 trial represents a replicate and a validation of the 2016 trial, and good correlations between them demonstrate that this is correct. 

We added suppl Fig 4 to show that both HS trials show comparable phenotypic correlations, which indicates sufficient data quality. 

L376 – Missing data rate less than 0.5 is very high. Usually a filter is applied around 0.8 to 0.9. Especially for Beadchip technology, where the genotyping error rate is usually very low. 

RESPONSE: Not sure if I understand that comment. We made a cutoff at about <=1% missing data rate (50 out of ~4800). A missing data rate of 0.9 would mean that nearly all the data is missing, one should be more stringent than that. None of this refers to a genotyping error rate as I understand it, rather to missing data points. Maybe the terms need to be clarified. 

L385 – Poor caption, information on the number of population genotypes, number of markers, difference between bars representing QTL and so on is lacking. 

improved

L399/401 – This belongs to the discussion. 

Sentence modified

Discussion 

The discussion, besides having several formatting errors, both in the text and in the references, has several statements that need citations. In addition, based on the complexity of the trials and the magnitude of the traits evaluated, many more things could be discussed and discussed. 

RESPONSE: Surely many more topics could be discussed, the described work touches on many interesting field. We hope we managed to make a good selection. 

L444/446 – The phrase is confused. 

revised

L447/451 - Provide the reference. 

Added

L454/455 - The statement is not entirely true. It is worth mentioning that although the controlled environment does not reflect the real conditions, the evaluation in a controlled environment for a study whose main objective is "Physiological and genetic characterization" would be ideal. Field conditions being the best for selecting superior strains. 

RESPONSE: That is correct for projects that are not directly interested in phlysiological effects in farmers’ fields. The aim of this project was to study field conditions, to generate knowledge that can be applied in improving smallholder farming systems. It’s a bit long to describe this in the title. We modified the discussion to reflect this better. 

L466/467 - Due to the difference in conducting the trials, it is not possible to compare HS2016 with 2017 and conclude that nighttime temperatures have major importance in heat stress. 

RESPONSE: We modified the sentence, adding a reference for the statement. The only known difference between the 2016 and 2017 trials is the climatic difference between the two years, meaning the difference in temperature. The observations may not be a proof, but they surely provide further evidence to the concept that night time temperatures are more important, which has previously been established in controlled condition experiments. 

L504/505 - Provide the reference. 

Modified and reference added

L512/513 - Provide the reference. 

RESPONSE: We would say that the notion that low grain quality reduces the price of the produce and thereby the farmers’ income is so basic that a reference for this is not required. This is occasionally stated in publications, e.g. by Ishimaru et al (2015) for rice or Kazai et al 2019 for bean, but those authors also do not give a reference for that. 

L515/516 - Incomplete and missing reference. 

RESPONSE: Again, the idea that molecular breeding tools are valuable for traits that are difficult to phenotype is so basic that adding a reference does not seem required. 

L525/526 - Repetitive, paragraph begins the same statement. 

Paragraphs was modified

L527 – Change “QTL” by “alleles”. 

changed

References 

L562 – Translate to English and update for 2018 (latest data available).

changed 

L604 – Correct the reference for “Blair MW, Hoyos A, Cajiao C, Kornegay J. Registration of Two Mid‐Altitude Climbing Bean Germplasm Lines with Yellow Grain Color, MAC56 and MAC57. J. Plant Reg. 2007;1: 143-144. doi.org/10.3198/jpr2006.09.0571crg”

corrected

Supporting information 

L711 – Change “y” by “and”.

corregido

Reviewer #2: 

On the whole the paper is well written and the work carried out flawlessly.

the few things I would like to point out are:

row 30 (Abstract) - it is not specified respect to what there was a decrease of 37% and 26% in the 2016 and 2017 seasons.

RESPONSE: The missing description was added.

row 90-91 (Introduction) - in these lines the citations have been indicated in a different form than the rest of the paper.

RESPONSE: this error was fixed

row 236 (Data Analysis) - I suggest starting the paragraph with "Phenotypic data analysis".

RESPONSE: The suggestion was followed.

row 237 (Data Analysis) - " HS2016 y NS2017" maybe "y" is a typing error or a Spanish residue.

RESPONSE: This was indeed a Spanish residue, and corrected.

row 319 - 327 (Result) - refers to GXE effects and after GxE ... maybe a typing error.

RESPONSE: typing error corrected

tabel 1-3 -there is a reference to NS2018, maybe is a typing error.

RESPONSE: The non-stress data set of Flowering Time actually dates from 2018. We failed to properly measure this trait in 2017. This is described in M&M under experimental locations. 

Furthermore, I suggest more details regarding the genetic mapping and identification of QTL,

the paper does not specify which "statistical strategy" is used through the QTL IciMapping software,

(for example the use of the "Inclusive composite interval mapping" method can be specified, or even the threshold

of statistical significance in QTL analysis).

RESPONSE: Information was added. 

Finally, I suggest to include in the discussions a reflection on the fact that this trial was carried out in the open field and in two different locations, and that this may affect the relationship between a phenotype and its association with heat stress conditions affecting the QTL mapping interpretation. This is because the different environmental conditions between the two locations were not considered in the paper except for the temperature. Perhaps specifying why this test was performed in the field and not in the greenhouse and consequently the relative advantages and disadvantages.

RESPONSE:

Thank you for your comments on the difficulty to perform heat stress trials in the field, due to the impossibility to create ideal control conditions. We extended our discussion of this topic at the beginning of the discussion section. A field trial cannot be performed at the same location with and without heat stress. This would only be possible in artificial controlled environments such as greenhouses, which do not fully represent conditions in farmers’ field. The aim of this project was to evaluate a situation as close to real production conditions as possible. We discuss the advantages and drawbacks of this approach.

---

## [Decision Letter · Decision Letter 1]

26 Mar 2021

Physiological and genetic characterization of heat stress effects in a common bean RIL population

PONE-D-20-34193R1

Dear Dr. Raatz,

We’re pleased to inform you that your manuscript has been judged scientifically suitable for publication and will be formally accepted for publication once it meets all outstanding technical requirements.

Kind regards,

Roberto Papa, PhD

Academic Editor

PLOS ONE

Additional Editor Comments (optional):

Reviewers' comments:

Reviewer's Responses to Questions

**Comments to the Author**

1. If the authors have adequately addressed your comments raised in a previous round of review and you feel that this manuscript is now acceptable for publication, you may indicate that here to bypass the “Comments to the Author” section, enter your conflict of interest statement in the “Confidential to Editor” section, and submit your "Accept" recommendation.

Reviewer #1: All comments have been addressed

Reviewer #2: All comments have been addressed

2. Is the manuscript technically sound, and do the data support the conclusions?

Reviewer #1: Yes

Reviewer #2: Yes

3. Has the statistical analysis been performed appropriately and rigorously? 

Reviewer #1: Yes

Reviewer #2: Yes

4. Have the authors made all data underlying the findings in their manuscript fully available?

Reviewer #1: Yes

Reviewer #2: Yes

5. Is the manuscript presented in an intelligible fashion and written in standard English?

Reviewer #1: Yes

Reviewer #2: Yes

6. Review Comments to the Author

Reviewer #1: I would like to thank the authors for their detailed responses toward my concerns and congratulate Bodo Raatz and your team for the work they have been doing in improving beans, especially for topics as important as drought tolerance and high temperature.

I believe that my biggest doubt about the study was related to the feasibility of the trials, however, the high correction between them demonstrated by the authors really shows that the increased noise in the lower quality trial is not masking the genetic effects.

Once again, congratulations on your study!

Reviewer #2: Following the corrections made by authors, I found the work complete and well articulated. In my opinion each comments has been satisfactorily addressed.

7. PLOS authors have the option to publish the peer review history of their article (what does this mean?). If published, this will include your full peer review and any attached files.

Reviewer #1: **Yes: **Caléo Panhoca Almeida

Reviewer #2: No

---

## [Editor Report · Acceptance letter]

12 Apr 2021

PONE-D-20-34193R1 

Physiological and genetic characterization of heat stress effects in a common bean RIL population 

Dear Dr. Raatz:

I'm pleased to inform you that your manuscript has been deemed suitable for publication in PLOS ONE. Congratulations! Your manuscript is now with our production department. 

Kind regards, 

on behalf of

Prof. Roberto Papa 

Academic Editor

PLOS ONE